# Longevity defined as top 10% survivors and beyond is transmitted as a quantitative genetic trait

Niels van den Berg [1,2,3], Mar Rodríguez-Girondo[4], Ingrid K. van Dijk[3], Rick J. Mourits[3], Kees Mandemakers[5], Angelique A.P.O. Janssens[3], Marian Beekman [1], Ken R. Smith[2] & P. Eline Slagboom[1,6]

Survival to extreme ages clusters within families. However, identifying genetic loci conferring longevity and low morbidity in such longevous families is challenging. There is debate concerning the survival percentile that best isolates the genetic component in longevity. Here, we use three-generational mortality data from two large datasets, UPDB (US) and LINKS (Netherlands). We study 20,360 unselected families containing index persons, their parents, siblings, spouses, and children, comprising 314,819 individuals. Our analyses provide strong evidence that longevity is transmitted as a quantitative genetic trait among survivors up to the top 10% of their birth cohort. We subsequently show a survival advantage, mounting to 31%, for individuals with top 10% surviving first and second-degree relatives in both databases and across generations, even in the presence of non-longevous parents. To guide future genetic studies, we suggest to base case selection on top 10% survivors of their birth cohort with equally long-lived family members.

[1] Department of Biomedical Data Sciences, Section of Molecular Epidemiology, Leiden University Medical Center, Albinusdreef 2, 2333 ZA Leiden, The Netherlands. [2] Department of Family and Consumer Studies, Population Sciences, Huntsman Cancer Institute, University of Utah, 225 S. 1400 E. Rm 228, Salt Lake City, UT, USA. [3] Radboud Group for Historical Demography and Family History, Radboud University, Erasmusplein 1, 6525 HT Nijmegen, The Netherlands. [4] Department of Biomedical Data Sciences, Section of Medical Statistics, Leiden University Medical Center, Albinusdreef 2, 2333 ZA Leiden, The Netherlands. [5] International Institute of Social History, Cruquiusweg 31, 1019 AT Amsterdam, The Netherlands. [6] Max Planck Institute for Biology of Ageing, Joseph-Stelzmann-Str. 9b, 50931 Cologne, Germany. These authors jointly supervised this work: Ken R. Smith, P. Eline Slagboom. Correspondence and requests for materials should be addressed to N.B. (email: n.m.a.van_den_berg@lumc.nl)

Human lifespan has a low heritability (12–25%)[1–4], whereas survival into extreme ages (longevity) clusters within families[5–9]. Studies showed that parents, siblings[5–7,9–12], and children[7,13–17] of longevous persons lived longer than first-degree relatives of non-longevous persons or population controls. In addition, members of these longevous families seem to delay or even escape age-related diseases[18–21] and in fact, healthy ageing in such families is marked by well attuned immune systems and good metabolic health[22–24]. Understanding the genetic factors influencing longevity may provide novel insights into the mechanisms that promote health and minimize disease risk[1,25]. Identifying longevity loci, however, has been challenging and only a handful of genetic variants have been shown to associate with longevity across multiple independent studies[25–32]. The most consistent evidence has been obtained for variants in APOE and FOXO3A genes[25–30,33] in either genome-wide association studies (GWAS) or candidate gene studies.

The lack of consistent findings in longevity studies hampers comparative research and may be explained by genetic and environmental heterogeneity on one hand and uncertainty in defining the longevity trait itself, as illustrated by the large variation of longevity definitions on the other hand[1,3,7,10,13–17,19,20,25–32,34–38]. Establishing a threshold that best isolates the genetic component of longevity and including mortality information of family members is important because the environmentally-related increase in lifespan over recent decennia has caused an increase in longevity phenocopies. As a result, genetic longevity studies generally focus on singletons (i.e., individuals without longevous family members), selected based on one generation of mortality data[27,28,31,32,39]. Here, we aim to establish the threshold for longevity in unselected (for survival) multigenerational families and determine the importance of longevous family members for case selection so that those insights can be used in genetic studies to identify novel longevity loci.

We use the data available in the Utah Population Database (UPDB, Utah) and the LINKing System for historical family reconstruction (LINKS, Zeeland) based on US and Dutch citizens, respectively. Zeeland was a region with difficult living conditions compared to Utah (see Methods section). In these datasets, we identify 20,360 three-generational families (F1–F3) containing index persons (IPs, F2), their parents (F1), siblings (F2), spouses (F2), and children (F3) comprising 314,819 persons in total. First, we examine the association between the survival, measured as age at death, of IPs (F2) and the number of parents (F1) and siblings (F2) belonging to the top 1–60% of their birth cohort, in a cumulative way (comparing mutually inclusive percentile groups). Second, we determine the survival percentile threshold that drives the cumulative effects as a criterion for defining human longevity by investigating IP (F2) survival when divided into mutually exclusive groups based on the longevity of their parents (F1) and siblings (F2). Third, we focus on the top 10% parents and siblings to investigate whether longevous and non-longevous parents, with increasing number of longevous siblings, transmit longevity to the IPs. Fourth, we confirm our findings in the next generation (F3) by examining the association between the survival, measured as age at death or last observation, of IPs' children (F3) and longevity of IPs (F2), their spouses (parents, F2) and siblings (aunts and uncles, F2). Finally, we explore potential environmental influences by studying spouses (F2) of longevous IPs (F2).

## Results

### Study population
We identified three generations of families in the UPDB and LINKS covering 10,246 and 10,114 families, respectively, who were centered around a single IP (F2) per family (Fig. 1). We

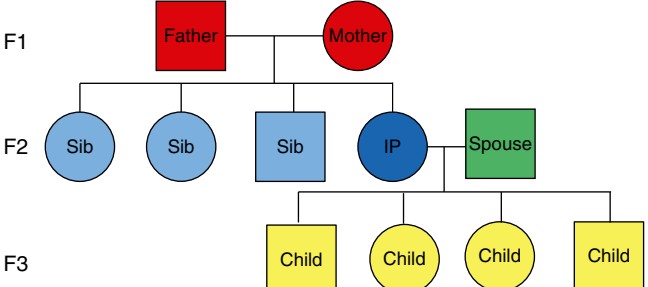

**Fig. 1** Conceptual pedigree of a 3 filial (F) generation family in the study. This figure represents a hypothetical family from the UPDB or LINKS covering 3 filial (F) generations. Circles represent women, squares represent men. Dark blue: index persons (F2), red: parents (F1), light blue: siblings of IP (F2), green: spouses of IP (F2), yellow: children of IP (F3). IP index person, Sib sibling, F filial

identified parents (F1, $N_{UPDB} = 20,492$ and $N_{LINKS} = 20,228$), siblings (F2, $N_{UPDB} = 54,144$ and $N_{LINKS} = 53,978$), spouses (F2, $N_{UPDB} = 11,230$ and $N_{LINKS} = 10,788$), and children (F3, $N_{UPDB} = 61,104$ and $N_{LINKS} = 62,495$) for all IPs in both datasets (Table 1). IPs were born between 1767 and 1902 in the UPDB, and between 1797 and 1902 in LINKS. In the UPDB, 51% of the IPs were female, compared to 53% in LINKS. The IPs' mean age at death was 70.88 (SD = 16.03) years in the UPDB and 63.86 (SD = 17.99) years in LINKS. No IPs were censored, as they were selected to have an available birth and death date. In addition, Supplementary Figure 1 shows the age at death distribution for the IPs in both datasets. In the following sections, we explore associations between IP survival and the number of 1–60% surviving parents and siblings in a cumulative analysis and subsequently identify in mutually exclusive IP groups the survival percentile threshold that drives the cumulative effect and demarcates longevity (see Methods section).

### IP survival advantage with top 1–60% parents and siblings
For the first examination of the association between the number of parents (1 or 2, F1) and siblings (1 or 2+, F2) and IP (F2) survival and to explore if a larger level of family aggregation, in terms of numbers of parents (F1) and siblings (F2), was more evident at extreme survival percentiles, we fitted Cox regressions for each subsequent survival percentile (1st to 60th percentile). Figure 2a, c shows that IPs with 1 parent belonging to the top 1–60%, had a survival advantage over IPs without a parent belonging to the top 1–60%. This was shown by the lowest observed statistically significant hazard ratio (HR) of 0.80 (95% $CI_{max-top 1\%} = 0.73–0.88$) in the UPDB and 0.74 (95% $CI_{max-top 1\%} = 0.65–0.85$) in LINKS where max refers to the age with the largest effect and CI to confidence interval. These HRs indicate a 20% and 26% lower hazard of dying respectively and from here we will refer to this as a 20% and 26% survival advantage. Having 2 parents belonging to the top 1–60% provides a stronger survival advantage to IPs ($HR_{max-top 2\%-UPDB} = 0.64$ (95%CI = 0.50–0.84) and $HR_{max-top 14\%-LINKS} = 0.72$ (95% CI = 0.65–0.80)), although Fig. 2 shows that the power to detect survival effects of IPs with 2 longevous parents up to the 10th percentile was weak for LINKS due to low group sizes.

The association of IP survival with longevous siblings is shown in Fig. 2b, d. The maximum statistically significant HRs for IPs with 1 longevous sibling were 0.76 (95% $CI_{max-top 1\%} = 0.62–0.92$) and 0.79 (95% $CI_{max-top 5\%} = 0.67–0.93$) in the UPDB and LINKS, respectively. For IPs with 2 or more longevous siblings, these HRs were 0.65 (95% $CI_{max-top 3\%-UPDB} = 0.51–0.84$) and 0.67 (95% $CI_{max-top 8\%-LINKS} = 0.50–0.90$). The slopes in Fig. 2a–d show a

**Table 1 Overview of UPDB and LINKS index persons and their first-degree relatives and spouses**

|  | Parents F1 | IPs F2 | Siblings F2 | Spouses F2 | Children F3 |
|---|---|---|---|---|---|
| **UPDB** | | | | | |
| Number, N | 20,492 | 10,246 | 54,144 | 11,230 | 61,104 |
| Deceased, N (%) | 19,191 (94) | 10,246 (100) | 45,701 (84) | 10,256 (91) | 54,076 (88) |
| Female, N (%) | 10,246 (50) | 5193 (51) | 26,159 (48) | 5742 (51) | 29,675 (49) |
| Range birth cohorts | 1753-1884 | 1767-1902 | 1756-1932 | 1768-1922 | 1792-1937 |
| Mean ad or al, years (SD) | 68.96 (16.10) | 70,88 (16.03) | 44.32 (33.60) | 69.11 (17.58) | 54.87 (32.09) |
| Mean ad, years (SD) | 70.05 (15.23) | 70.88 (16.03) | 49.98 (32.45) | 70.75 (16.43) | 57.98 (31.28) |
| Missing age, N (%) | 403 (2) | 0 (0) | 799 (1) | 345 (3) | 306 (1) |
| Censored, N (%) | 898 (4) | 0 (0) | 7644 (14) | 629 (6) | 6722 (11) |
| **LINKS** | | | | | |
| Number, N | 20,228 | 10,114 | 53,978 | 10,788 | 62,495 |
| Deceased, N (%) | 15,536 (77) | 10,114 (100) | 40,093 (74) | 8819 (82) | 43,896 (70) |
| Female, N (%) | 10,114 (50) | 5338 (53) | 25,946 (48) | 5193 (48) | 30,347 (49) |
| Range birth cohorts | 1740-1877 | 1797-1902 | 1796-1916 | 1775-1907 | 1818-1952 |
| Mean ad or al, years (SD) | 54.65 (20.66) | 63.86 (17.99) | 20.84 (27.99) | 59.04 (21.23) | 24.86 (30.06) |
| Mean ad, years (SD) | 62.64 (16.15) | 63.86 (17.99) | 23.94 (30.76) | 65.70 (17.20) | 29.59 (33.63) |
| Missing age, N (%) | 49 (<1) | 0 (0) | 14 (<1) | 27 (<1) | 21 (<1) |
| Censored, N (%) | 4643 (23) | 0 (0) | 13,878 (26) | 1942 (18) | 18,578 (30) |

*ad* age at death, *al* age at last observation, *IPs* index persons. Missing age means that we have no observations at all

slight increase of IP survival advantage with the increase in percentile score. For example, IPs with parents with the best survival (the left-most end of the $x$-axis) had lower hazard rates than IPs with the least survival (the right-most end of the $x$-axis). We conclude that IP survival when expressed in HRs, both in the UPDB and LINKS, increased with the number of longevous parents, with the number of longevous siblings and, though modestly, with the increase of parent and sibling survival percentile scores as observed in Fig. 2.

**Top 10–15% surviving family members demarcates longevity.** To determine the survival percentile threshold that drove the survival advantage of IPs (F2) with the number of top 1–60% parents (F1), as shown in Fig. 2, we constructed 6 mutually exclusive IP (F2) groups (g) based on the survival percentiles of F1 parents (g1 = [≥0th & ≤1th percentile], g2 = [≥1th & ≤5th percentile], g3 = [≥5th & ≤10th percentile], g4 = [≥10th & ≤15th percentile], g5 = [≥15th & ≤20th percentile], g6 = [≥20th & ≤100th percentile], see Methods section) and compared groups 1–5 with group 6. Figure 3a, b shows the HRs of IP groups for the UPDB and LINKS and is supplemented by the IP age at death and survival percentile variation, as depicted in Supplementary Figure 2. Figure 3a, b illustrates that IPs in groups 1, 2, 3, and 4 had a significant survival advantage compared to group 6, with the lowest HR for group 1 in both the UPDB and LINKS ($\text{HR}_{\text{max-UPDB}} = 0.76$ (95% CI = 0.67–0.86) and $\text{HR}_{\text{max-LINKS}} = 0.72$ (95% CI = 0.60–0.86)). Group 5 did not statistically differ from group 6 ($\text{HR}_{\text{group5-UPDB}} = 1$ (95% CI = 0.91–1.10) and $\text{HR}_{\text{group5-LINKS}} = 0.96$ (95% CI = 0.87–1.05)) and thus, these effects indicate that the top 15% surviving parents drove the association with the survival advantage of IPs as shown in Fig. 2.

In the same way, we investigated the association of IPs' (F2) survival with that of siblings (F2). Figure 3c, d shows a survival advantage of IPs in UPDB group 1–3 and LINKS group 2 and 3 as compared to group 6 with the lowest HR for group 1 (UPDB) and group 2 (LINKS) ($\text{HR}_{\text{group1-UPDB}} = 0.70$ (95% CI = 0.59–0.85) and $\text{HR}_{\text{group2-LINKS}} = 0.77$ (95% CI = 0.65–0.92)), respectively. Groups 4 and 5 did not significantly differ from group 6 ($\text{HR}_{\text{group4-UPDB}} = 0.99$ (95% CI = 0.88–1.12) and $\text{HR}_{\text{group4-LINKS}} = 0.86$ (95% CI = 0.73–1.02)), which indicated that both in the UPDB and LINKS the top 10% surviving siblings drove the association with the survival advantage of IPs as shown in Fig. 2.

Based on the results presented in the cumulative and mutually exclusive group analyses, we focused on the top 10% surviving family members because the mutually exclusive group analysis (analysis 2, Fig. 3) indicated longevity effects up to the top 10% and 15% for siblings and parents, respectively. Using the top 10% is consistent between the two groups and is a conservative choice. Furthermore, the cumulative analysis (analysis 1, Fig. 2) indicated that the top 10% was a reasonable trade-off between effect size and group size (power) within and between the UPDB and LINKS. Hence, we explored the familial clustering of longevity and the influence of covariates for the top 10% surviving parents and siblings and verified all results in the subsequent generation (F3). Next to the top 10% we also conducted our analyses on the top 5% which are illustrated in Supplementary Figures 3–5 and Supplementary Tables 1–5.

**Additive association between 10% surviving relatives and IPs.** Figure 2e–h shows the cumulative hazard (CH) curves for IPs (F2) with 0, 1 and 2 or more, or exactly 2 parents/siblings (F1/F2) belonging to the top 10% of their birth cohorts and we show Kaplan–Meier and Nelson–Aalen baseline measures in Supplementary Figure 6. Both in the UPDB and LINKS, the survival advantage associated with the number of top 10% siblings appeared to start during the beginning (45 years in LINKS) and end (65 years in the UPDB) of the mid-life period. In both the UPDB and LINKS, the survival advantage of IPs with the number of top 10% parents started at the age of 40 years. It should be noted that early life effects could not be tested for, because IPs were selected on having a child for the construction of three generation families.

Table 2 accompanies Fig. 2e–h by showing the HRs for the number of top 10% parents (F1) and siblings (F2) and for the covariates we used to adjust the analyses. IPs with 1 top 10% parent had a maximum survival advantage of 12% and 18% compared to IPs without such a parent ($\text{HR} = 0.88_{\text{max-UPDB}}$ (95% CI = 0.83–0.92) and $\text{HR} = 0.82_{\text{max-LINKS}}$ (95% CI = 0.78–0.86)). The maximum statistically significant survival advantage for IPs with 2 top 10% parents was 27% and 31% ($\text{HR}_{\text{max-UPDB}} = 0.73$ (95% CI = 0.65–0.83) and $\text{HR}_{\text{max-LINKS}} = 0.69$ (95% CI = 0.58–0.82)). The maximum statistically significant HR for having 1 top 10% sibling was 0.82 (95% $\text{CI}_{\text{UPDB}} = 0.76$–0.90) and 0.82 (95% $\text{CI}_{\text{LINKS}} = 0.73$–0.93). For 2+ top 10% siblings the HR was

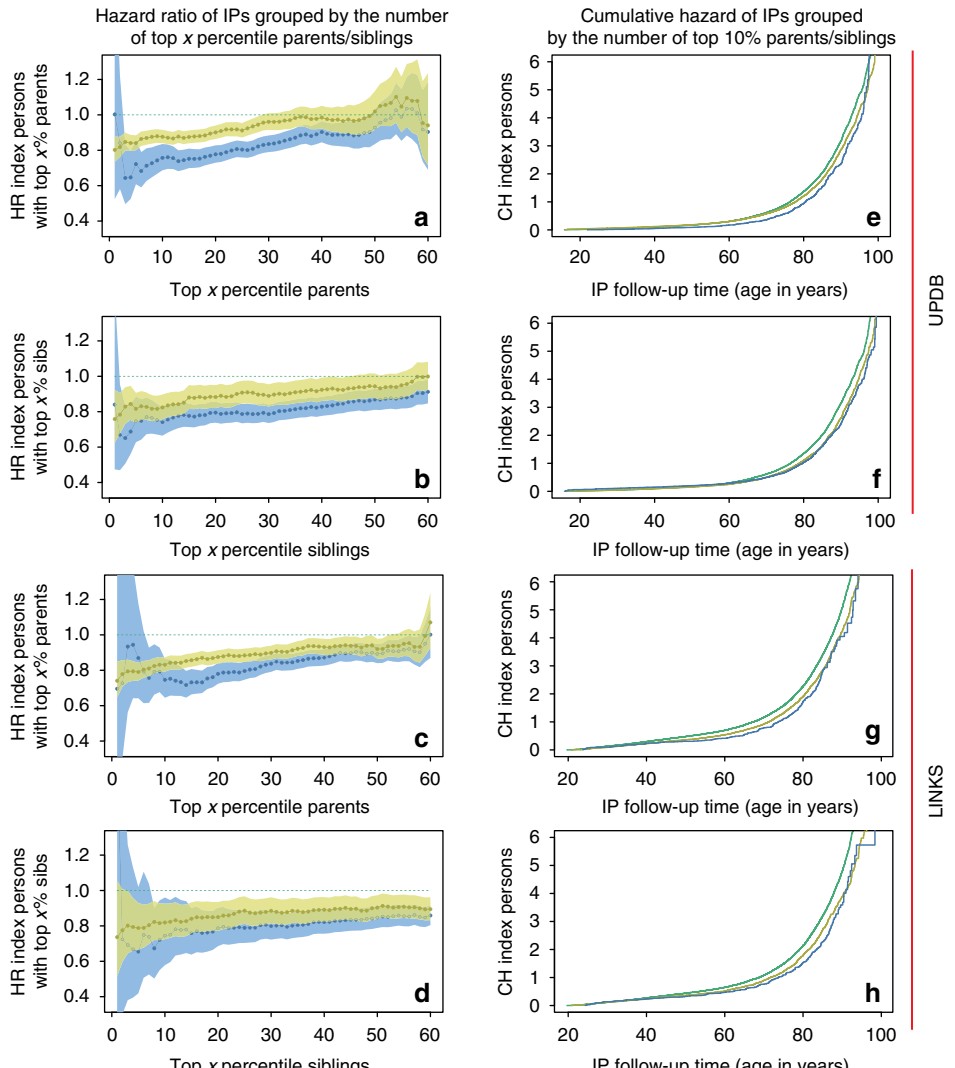

**Fig. 2** Survival of IPs with relatives belonging to the 1st until 60th percentile survivors of their birth cohort. This figure depicts the hazard ratio (HR) for IPs (left column, panels **a**–**d**) with 1 and 2 parents (panels **a** and **c**) or 1 and 2+ siblings (panels **b** and **d**) belonging to the top $x$ percentile ($x = 1,2,3, ..., 60$) survivors of their birth cohort. The percentile groups ($x$-axis) are mutually inclusive, meaning that a first-degree family member who belonged to the top 1% also belonged to the top 5%, etc. The figure also depicts the cumulative hazard (CH) for index persons (IPs, right column, panels **e**–**h**) with 1 and 2 parents (panels **e** and **g**) or 1 and 2+ siblings (panels **f** and **h**) who belong to the top 10%. Green (dotted) lines present the reference group of 0 top $x$ percentile parents or siblings, yellow lines represent 1 top $x$ percentile parents or siblings, blue lines represent 2 or 2+ top $x$ percentile parents or siblings. Left column: $x$-axes represent the top $x$ birth cohort-based survival percentile, the $y$-axes represent the hazard ratio (HR) of dying for IPs having 1 and 2 or 2+ top $x$ percentile parents or siblings compared to having 0 top $x$ percentile parents or siblings. Right column: $x$-axes represent IP years of survival, $y$-axes represent the IPs' cumulative hazard of dying while having 1 and 2 or 2+ top 10th percentile parents or siblings compared to having 0 top 10th percentile parents or siblings. All estimates are adjusted for religion (UPDB only), sibship size, birth cohort, sex, socio-economic status, mother's age at birth, birth order, birth intervals, twin birth, and number of top 10% parents or number of top 10% siblings for the sibling and parent analyses, respectively. Error bars represent confidence intervals

0.74 (95% $CI_{UPDB}$ = 0.66–0.82) and 0.75 (95% $CI_{LINKS}$ = 0.58–0.96). The survival advantage of IPs with 1 and 2 or more, or exactly 2 top 10% siblings and parents respectively was independent of covariates such as sibship size and religion (LDS church affiliation, Table 2 and Supplementary Table 6 and 7). Religious IPs from Utah had a lower HR than non-religious persons ($HR_{UPDB}$ = 0.73 (95% CI = 0.65–0.81)) and in the UPDB we observed that sibship size had a small influence on the survival of IPs ($HR_{UPDB}$ = 1.01 (95% CI = 1.00–1.02) whereas in LINKS, sibship size had no significant effect $HR_{LINKS}$ = 1.01 (95% CI = 1.00–1.02)). The survival of IPs increased with the increase of birth cohort ($HR_{UPDB \ and \ LINKS}$ = 0.99 (95% CI = [>0.99 to <1.00])) and women had a better survival than men in the UPDB

($HR_{UPDB}$ = 0.71 (95% CI = 0.67–0.76)), but not in LINKS ($HR_{LINKS}$ = 1.01 (95% CI = 0.96–1.06)). Furthermore, in Utah, high socio-economic status IPs outlived low socio-economic status IPs whereas this was not the case in LINKS. The association between the number of longevous parents/siblings and the survival of IPs were independent of each other and no other statistically significant effect was observed for having both longevous parents and siblings. Moreover, the number of longevous siblings showed a strong association with the survival of IPs when both parents were non-longevous. The HR for 1 longevous sibling was 0.85 (95% CI = 0.79–0.91) and the HR for 2 or more longevous siblings was 0.78 (95% CI = 0.67–0.90) in the UPDB. The HR for 1 longevous sibling was 0.78 (95%

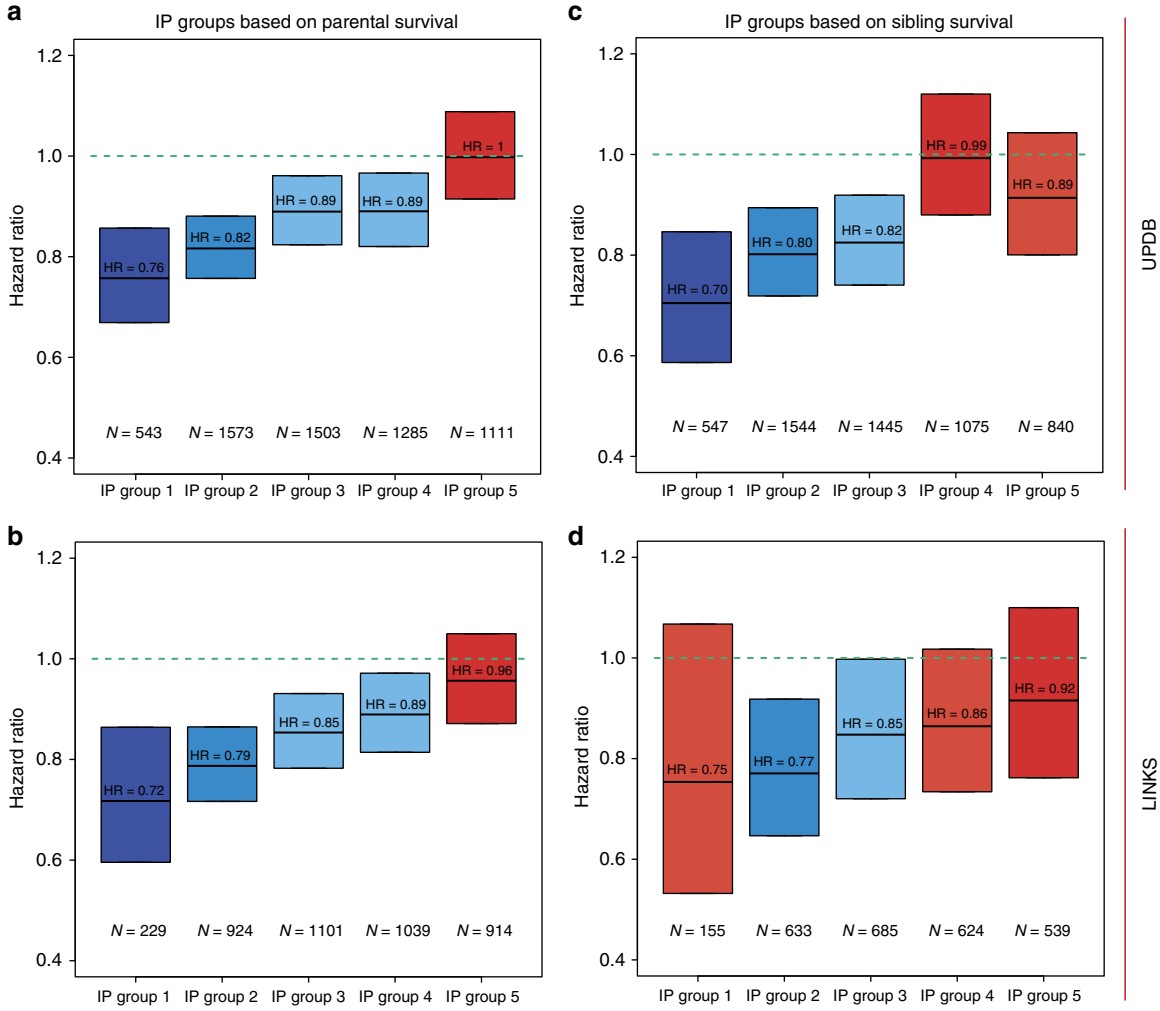

**Fig. 3** Hazard ratio for IPs grouped by their relatives' survival in mutually exclusive groups. Parent and sibling groups: group 1 = IPs of whom the longest lived parent/sibling belonged to the [≥0th & ≤1th percentile] of their birth cohort, group 2 = IPs of whom the longest lived parent/sibling belonged to the [≥1th & ≤5th percentile], group 3 = IPs of whom the longest lived parent/sibling belonged to the [≥5th & ≤10th percentile], group 4 = IPs of whom the longest lived parent/sibling belonged to the [≥10th & ≤15th percentile], group 5 = IPs of whom the longest lived parent/sibling belonged to the [≥15th & ≤20th percentile], group 6 = IPs of whom the longest lived parent/sibling belonged to the [≥20th & ≤100th percentile]. The left column (panels **a** and **b**) shows the HRs of IP groups 1–5 compared to group 6 and depicts a parental grouping. The right column (panels **c** and **d**) shows the HRs of IP groups 1–5 compared to group 6 and depicts a sibling grouping. Groups were colored by the extremity of the HR. The darker the blue the stronger the survival benefit, the darker the red, the weaker the survival benefit and the effect was not significant with the red colors. The green lines represent the reference category, which is group 6. $N_{green\ line}$ at the top-right = 4759, $N_{green\ line}$ at the top-left = 4227, $N_{green\ line}$ at the bottom-right = 7477, $N_{green\ line}$ at the bottom-left = 5907. All estimates are adjusted for religion (UPDB only), sibship size, birth cohort, sex, socio-economic status, mother's age at birth, birth order, birth intervals, twin birth, and number of top 10% parents or number of top 10% siblings for the sibling and parent analyses, respectively. Error bars represent confidence intervals

CI = 0.72–0.85) and the HR for 2 or more longevous siblings was 0.72 (95% CI = 0.53–0.99) in LINKS (Supplementary Table 8). In a final step, we observed no evidence that the association of IP survival and parental longevity depended on maternal or paternal effects, for example through transmission preferentially via the mother or father (Supplementary Table 9). Likewise, the association of IP survival and parental longevity did not depend on the sex of the IPs, meaning that this association was equal for sons and daughters (Supplementary Figure 7 and Supplementary Table 10).

**Survival advantage for children with longevous relatives.** We explored the robustness of our findings in F1 and F2 by examining the association between the longevity of IPs (F2), their spouses (F2)

and siblings (F2) and the survival of IPs' children (F3). We investigated whether longevity was transmitted from IPs (F2) to their children (F3) and if the children (F3) with longevous aunts and uncles (siblings of the IPs, F2) had a survival advantage compared to children (F3) without longevous aunts and uncles (F2). To test this, we fitted Cox regressions, with a random effect (frailty) to adjust for within-family relations of the F3 children. Table 3 shows that children of a top 10% surviving IP had a HR of 0.86 (95% $CI_{UPDB}$ = 0.84–0.89) in the UPDB and 0.85 in LINKS (95% $CI_{LINKS}$ = 0.82–0.88) compared to children without a top 10% IP. Moreover, results indicated that children with two top 10% parents (IPs and spouses) had a HR of 0.77 (95% $CI_{UPDB}$ = 0.72–0.82) in the UPDB and 0.77 (95% $CI_{LINKS}$ = 0.71–0.84) in LINKS. Similar to the IPs, we observed (1) that the survival of children did not depend on maternal or paternal effects (Supplementary Table 9) and (2)

**Table 2 Survival analysis for IPs with top 10% parents and siblings**

| | UPDB | | | LINKS | | |
|---|---|---|---|---|---|---|
| | N (mean) | HR (95% CI) | p-Value | N (mean) | HR (95% CI) | p-Value |
| Top 10% parents (F1) | | | | | | |
| 0 (ref) | 6640 (0.65) | | | 7861 (0.78) | | |
| 1 | 3167 (0.31) | 0.88 (0.83-0.92) | $2.94*10^{-7}$ | 2096 (0.20) | 0.82 (0.78-0.86) | $1.27*10^{-13}$ |
| 2 | 439 (0.4) | 0.73 (0.65-0.83) | $4.38*10^{-7}$ | 184 (0.2) | 0.69 (0.58-0.82) | $1.91*10^{-5}$ |
| Top 10% sibs (F2) | | | | | | |
| 0 (ref) | 6720 (0.66) | | | 8644 (0.85) | | |
| 1 | 2495 (0.24) | 0.82 (0.76-0.90) | $6.85*10^{-6}$ | 1256 (0.13) | 0.82 (0.73-0.93) | $1.38*10^{-3}$ |
| 2+ | 1031 (0.10) | 0.74 (0.66-0.82) | $4.15*10^{-8}$ | 214 (0.2) | 0.75 (0.58-0.96) | $2.30*10^{-2}$ |
| LDS (F2) | | | | | | |
| 0—non-religious (ref) | 2753 (0.27) | | | | | |
| 1—baptized | 512 (0.05) | 0.73 (0.65-0.81) | $1.49*10^{-8}$ | NA | NA | NA |
| 2—baptized + endowment | 6736 (0.66) | 0.80 (0.76-0.85) | $2.33*10^{-15}$ | NA | NA | NA |
| 3—missing | 245 (0.02) | 0.85 (0.73-0.99) | $4.24*10^{-2}$ | NA | NA | NA |
| Sibship size (F2) | 10,246 (6.28) | 1.01 (1.00-1.02) | $2.23*10^{-2}$ | 10,114 (6.34) | 1.01 (1.00-1.02) | $9.02*10^{-2}$ |
| Birth cohort, years (F2) | 10,246 (1868) | 0.99 (>0.99 to <1.00) | $2.66*10^{-09}$ | 10,114 (1835) | 0.99 (>0.99 to <1.00) | $<1.00*10^{-15}$ |
| Sex (F2) | | | | | | |
| Man (ref) | 5053 (0.49) | | | 4776 (0.48) | | |
| Women | 5193 (0.51) | 0.71 (0.67-0.76) | $<1.00*10^{-15}$ | 5338 (0.52) | 1.01 (0.96-1.06) | $7.53*10^{-1}$ |
| SES—OCC_1950 (F2) | | | | | | |
| 0—High (ref) | 315 (0.03) | | | 67 (0.01) | | |
| 1 | 1482 (0.14) | 1.16 (1.01-1.34) | $3.95*10^{-2}$ | 645 (0.06) | 0.88 (0.68-1.14) | $3.42*10^{-1}$ |
| 2 | 400 (0.04) | 1.19 (1.00-1.40) | $4.95*10^{-2}$ | 536 (0.05) | 0.97 (0.75-1.27) | $8.42*10^{-1}$ |
| 3 | 352 (0.03) | 1.24 (1.05-1.48) | $1.38*10^{-2}$ | 62 (0.01) | 0.76 (0.53-1.10) | $1.45*10^{-1}$ |
| 4 | 187 (0.02) | 1.14 (0.93-1.40) | $2.09*10^{-1}$ | 71 (0.01) | 0.99 (0.70-1.40) | $9.41*10^{-1}$ |
| 5 | 891 (0.09) | 1.31 (1.13-1.52) | $4.22*10^{-4}$ | 733 (0.07) | 0.80 (0.62-1.04) | $9.19*10^{-2}$ |
| 6 | 668 (0.07) | 1.34 (1.15-1.56) | $2.13*10^{-4}$ | 311 (0.03) | 0.86 (0.65-1.13) | $2.71*10^{-1}$ |
| 7 | 522 (0.05) | 1.27 (1.08-1.50) | $4.14*10^{-3}$ | 759 (0.08) | 0.84 (0.65-1.10) | $2.01*10^{-1}$ |
| 8 | 168 (0.02) | 1.21 (0.97-1.50) | $8.91*10^{-2}$ | 574 (0.06) | 0.85 (0.65-1.11) | $2.35*10^{-1}$ |
| 9—Low | 562 (0.05) | 1.48 (1.26-1.73) | $1.70*10^{-6}$ | 3656 (0.36) | 0.83 (0.65-1.07) | $1.56*10^{-1}$ |
| 999—missing | 4699 (0.46) | 1.61 (1.40-1.84) | $9.54*10^{-12}$ | 2700 (0.26) | 0.93 (0.72-1.20) | $5.95*10^{-1}$ |
| Log likelihood | −60,719 | | | −72,239 | | |

Table corresponds to the CH curves in the top and bottom right panel of Fig. 2. Means represent a mean for a continuous variable and a proportion for a categorical variable. Additional covariates are age of mom at birth, birth order, birth intervals (in years), twin birth. When the p-value was lower than 1.00e−15 we indicated the p-value as <1.00*10−15. *LDS* the church of Jesus Christ of latter-day saints (Mormon church), *SES* socio-economic status, *OCC* occupational coding scheme of 1950, *CI* confidence interval, *CH* cumulative hazard. p-Values are estimated with Cox regression

**Table 3 Frailty survival analysis for children of IP's with top 10% IPs and aunts and uncles**

| | UPDB | | | LINKS | | |
|---|---|---|---|---|---|---|
| | N (mean) | HR (95% CI) | p-Value | N (mean) | HR (95% CI) | p-Value |
| Top 10% IP (F2) | | | | | | |
| 0 non-LL (ref.) | 48,619 (0.80) | | | 53,378 (0.85) | | |
| 1 LL | 12,179 (0.20) | 0.86 (0.84-0.89) | $<1.00*10^{-15}$ | 9096 (0.15) | 0.85 (0.82-0.88) | $<1.00*10^{-15}$ |
| Top 10% aunts and uncles (F2) | | | | | | |
| 0 (ref.) | 39,474 (0.65) | | | 53,228 (0.85) | | |
| 1 | 15,134 (0.25) | 0.96 (0.93-0.99) | $3.19*10^{-3}$ | 7817 (0.12) | 0.96 (0.92-0.99) | $1.90*10^{-2}$ |
| 2+ | 6190 (0.10) | 0.92 (0.88-0.96) | $4.33*10^{-5}$ | 1429 (0.3) | 0.84 (0.78-0.92) | $5.47*10^{-5}$ |
| Sibshipsize (F3) | 60,798 (8.89) | 1.02 (1.01-1.02) | $<1.00*10^{-15}$ | 62,474 (8.52) | 1.00 (>0.99 to <1.00) | $7.87*10^{-1}$ |
| Birth year (F3) | 60,798 (1892) | 0.99 (>0.99 to <1.00) | $<1.00*10^{-15}$ | 62,474 (1867) | 0.99 (>0.99 to <1.00) | $2.37*10^{-11}$ |
| Sex (F3) | | | | | | |
| Man (ref.) | 31,258 (0.51) | | | 32,136 (0.52) | | |
| Women | 29,540 (0.49) | 0.62 (0.60-0.63) | $<1.00*10^{-15}$ | 30,338 (0.48) | 0.64 (0.63-0.66) | $<1.00*10^{-15}$ |
| Famid intercept (variance) | 60,798 (1.00) | 0.34 (0.11) | | 62,474 (1.00) | 0.34 (0.11) | |
| BIC | 60,798 (1.00) | −23,756.54 | | 62,474 (1.00) | −21,477.23 | |

Additional covariates are birth order, birth intervals (years), age of mom at birth. Religion, socio-economic status, twin birth have been stratified. When the p-value was lower than 1.00e−15 we indicated the p-value as <1.00e−15. *BIC* Bayesian Information Criterion, *Famid* family identifier, *CI* confidence interval, *LL* long lived. p-Values are estimated with Cox regression

that the association between parents and offspring was equal for sons and daughters (Supplementary Table 10).

Children with 1 or more top 10% aunts or uncles had a 4–16% survival advantage compared to children without such aunts or uncles ($HR_{min-UPDB} = 0.96$ (95% CI = 0.93–0.99) and $HR_{max-LINKS} = 0.84$ (95% CI = 0.78–0.92)), and this effect was independent of having a top 10% parent (either the IP or the IP's spouse). A stratified analysis showed that the survival benefit for children with the number of top 10% aunts and uncles was still strongly present when the IP and the IP's spouse were non-longevous ($HR_{min-UPDB-1\ aunt/uncle} = 0.96$ (95% CI = 0.93–0.99) and $HR_{max-LINKS-2+\ aunts/uncles} = 0.81$ (95% CI = 0.73–0.90)) (Supplementary Table 11). Lastly, Supplementary Figure 8 shows that the survival benefit for children of a longevous IP and a longevous IP with a longevous spouse (i.e., 1 or 2 longevous parents) started from birth (LINKS) and very early in life (UPDB).

**Spouses live longer in Zeeland but not in Utah.** Familial clustering of longevity may depend on (later life) shared environmental effects which could also provide survival benefits to the spouses (F2) of longevous IPs (F2). Hence, we divided the spouses (F2) into mutually exclusive groups according to the survival percentiles of the IPs (see Methods). Figure 4a, c shows that none of the spouse groups in the UPDB differed from reference group 6 or from any of the other groups, indicating no survival benefit for spouses. In LINKS (Fig. 4b, d), spouses of IPs with the highest survival percentile (group 2) had a 14% ($HR_{group2-LINKS} = 0.86$ (95% CI = 0.78–0.94)) survival advantage compared to group 6 spouses. This survival advantage was similar for spouses of IPs in groups 3, 4, and 5 ($HR_{group3-LINKS} = 0.86$ (95% CI = 0.80–0.94); $HR_{group4-LINKS} = 0.92$ (95% CI = 0.85–0.99); $HR_{group5-LINKS} = 0.86$ (95% CI = 0.79–0.93)). For group 1, the effect was comparable but not significant ($HR_{group1-LINKS} = 0.85$ (95% CI = 0.70–1.04)), the test in group 4 did not meet Bonferroni correction for multiple testing.

## Discussion

Human longevity clusters within specific families. Insight into this clustering is important, especially to improve our understanding of genetic and environmental factors driving healthy aging and longevity. The analyses of the UPDB and LINKS datasets, which cover different environmental circumstances, provide strong evidence that for longevous (up to the top 10%) survivors and their families, longevity is transmitted as a quantitative genetic trait, regardless of parental and offspring sex. The main observations supporting this notion are (1) in both datasets the survival of F2 IPs, and their F3 children, increased with each additional longevous parent (F1 and F2) and sibling (F2); (2) in both datasets the survival of IPs (F2) increased with the number of longevous siblings (F2) in the absence of longevous parents (F1) and likewise the survival of IPs' children (F3) increased with the number of longevous aunts and uncles in the absence of longevous parents. Finally, (3) both datasets indicate an absence of a sex-specific pattern.

Previous studies of smaller sample size than the current study, usually focusing on two generations of selected data (for mortality or geographical locations) identified (1) an increase in the heritability of lifespan with parental age[8,40,41] and showed high recurrence risks between parental and offspring or sibling longevity. Thus, providing indications that the heritability of longevity may be stronger than that of lifespan[5,7,13,14,42], (2) that sibling relative risks beyond the top 5% survivors might not increase in a linear fashion[12] and that this non-linearity may indicate the existence of a longevity threshold[43], and (3) longevity recurrence risks for siblings or parents of selected longevous individuals[5–7,9–12] and

showed increased survival probabilities and longevity recurrence risks for children of longevous parents[7,13–17].

Here we used two unique, large three-generational datasets (314,819 individuals in 20,360 families) which were unselected for survival and cover multiple geographical areas. We utilized these datasets to robustly identify a longevity threshold by showing that the association between IP survival and the survival of parents and siblings was not linear but in fact was driven by the oldest, up to the top 10%, surviving parents and siblings. We further showed that the survival of F2 IPs (and their F3 children) increased with each additional longevous parent (F1 and F2), and sibling (F2). We extended these analyses by showing that the survival for children of IPs increased with each additional longevous aunt or uncle. We also extended the analyses by investigating the association between IP survival and sibling longevity, and between the survival of IPs' children and the longevity of their aunts or uncles in the absence of longevous parents.

Longevity was transmitted even if parents themselves did not become longevous, which supports the notion that a beneficial genetic component was transmitted. Likewise, the identified associations are additive in the sense that an increase in the number of parents, siblings, or aunts and uncles is associated with an increase in the survival of IPs and the children of IPs. This additive pattern is not necessarily expected if the findings are due to other, non-genetic, factors that cluster within families (for example wealth). This evidence is strengthened by the fact that similar additive associations were identified for IPs and children of IPs without longevous parents but with longevous siblings or aunts and uncles (where the latter generally share less environmental influences with the IPs). Further evidence for the transmission of a genetic component was shown by the fact that none of the tested environmental confounders affected the associations between parental/sibling longevity and IP/children survival, as will be discussed further on in the Discussion section. In addition, the fact that we observed very similar results between the two databases, which cover populations with vastly different environmentally-related mortality regimes, significantly adds to the generalizability of our observations regarding the associations between parental/sibling longevity and IP (F2) and children (F3) survival.

We showed that spouses (F2) who married longevous IPs (F2) did not live significantly longer than spouses (F2) who married a non-longevous IP (F2) in the UPDB while they did in LINKS. Previous studies showed inconclusive results regarding a possible survival benefit for spouses of longevous persons[6,7,9,44,45]. In the Long Life Family study, Pedersen et al.[6] identified a survival benefit for spouses of longevous siblings. The authors compared the spouses to sex and birth cohort matched controls and suggest assortative mating as an explanation for the observed survival benefit of the spouses[6]. A Quebec study, focused on the spouses of 806 centenarians, also reported a survival benefit[44] and a study of Southern Italy demonstrated that male nonagenarians outlived their spouses, whereas this was not the case for female nonagenarians[45]. A recent study showed that the spouses of 944 nonagenarians had no survival benefit but a life-long sustained survival pattern similar to the general population[9]. An explanation for the difference between the UPDB and LINKS datasets may possibly be that Zeeland had a higher level of relatedness than in Utah. Zeeland had poor living conditions[46] and was characterized by out-migration to other provinces or abroad, but limited mobility within the province to other places[47]. Utah at that time had better living conditions[48] with continuous streams of freshly incoming migrants, ensuring a steady influx of new genes[49], creating high genetic diversity. Hence, it could be that in Zeeland, spouses and IPs were often related to each other and thus shared some of the genetic component contributing to longevity.

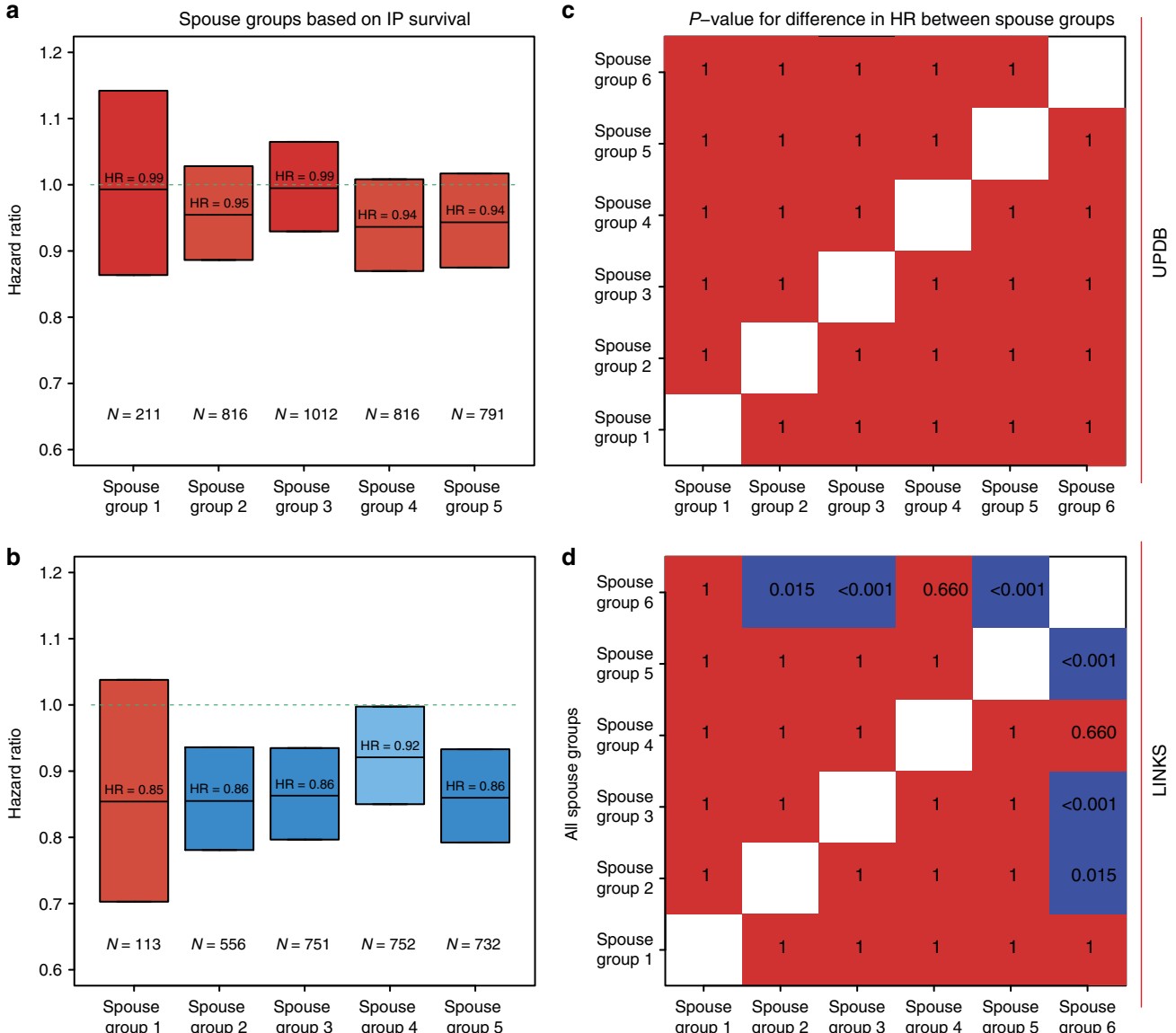

**Fig. 4** Hazard ratio for spouses grouped by IP survival in mutual exclusive groups. Spouse groups: group 1 = spouses of whom the IP belonged to the [≥0th & ≤1th percentile] of their birth cohort, group 2 = spouses of whom the IP belonged to the [≥1th & ≤5th percentile] of their birth cohort, group 3 = spouses of whom the IP belonged to the [≥5th & ≤10th percentile] of their birth cohort, group 4 = spouses of whom the IP belonged to the [≥10th & ≤15th percentile] of their birth cohort, group 5 = spouses of whom the IP belonged to the [≥15th & ≤20th percentile] of their birth cohort, group 6 = spouses of whom the IP belonged to the [≥20th & ≤100th percentile] of their birth cohort. The left column (panels **a** and **b**) shows the HRs of groups 1–5 compared to group 6. Groups were colored by the extremity of the HR. The darker the blue the stronger the survival benefit, the darker the red, the weaker the survival benefit and the effect was not significant with the red colors. The green lines represent the reference category, which is group 6. $N_{green\ line}$ at the top-left = 8065, $N_{green\ line}$ at the bottom-left = 7887. The right column (panels **c** and **d**) represents a post-hoc test of all groups and illustrates the p-values for the differences in HR between the spouse groups. p-Values are estimated with Cox regression. Blue color indicates a statistically significant effect after Bonferroni correction, red color indicates a non-statistically significant effect after Bonferroni correction. All estimates are adjusted for religion (UPDB only), sibship size, birth cohort, sex, socio-economic status, mother's age at birth, birth order, birth intervals, and twin birth. Error bars represent confidence intervals

Unlike observations we previously made in the Leiden Longevity Study (LLS)[9] concerning maternal effects on longevity in the generation of the nonagenarians and their parents, we did not observe evidence for a stronger transmission from either parent to the IPs (F1 to F2), or from IPs to their children (F2 to F3) in our current study. We cannot draw final conclusions on this aspect because for the F1–F2 transmission we may have missed parental influences on early life mortality since IPs were selected for having survived to an age at which they had one child.

However, we did capture early life mortality for F2–F3 but in those generations the selection pressure on child mortality was already slightly decreasing[50]. In the same way, we observed no differential association between parental longevity and the survival of sons or daughters (F2 and F3). This equal distribution is in line with our observations in the LLS[9], but so far, the literature was less conclusive on this point[15–17].

In all our analyses, except for the spouse analysis, we adjusted for religion (UPDB only), sibship size, birth cohort, sex,

socio-economic status, mother's age at birth, birth order, birth intervals, and twin birth. Some of these biological, social, and demographic factors associated with the mortality of IPs (F2) and their children (F3). Nevertheless, these covariates neither confounded the association between parental (F1) and sibling (F2) longevity and IP (F2) survival, nor that between IP (F2) and spouse (F2) longevity and their children's (F3) survival or between longevity of aunts and uncles (F2) and the survival of IPs' children (F3). This is in line with previous studies showing only a minor[51] or no[52] influence of environmental covariates on the association between parental longevity and offspring survival. It was also shown that a range of early life factors, such as farm ownership, parental literacy, and parental occupation did not affect the association between parental and offspring mortality[52]. We, however, cannot completely rule out that other, unobserved non-genetic familial effects may affect our results. Furthermore, using either Swedish or Dutch lifetables to determine survival percentiles was quite strict for Zeeland because of the hazardous environment[46]. As a result, the number of longevous persons was quite low in LINKS relative to the UPDB. Although the IPs were randomly selected, we could not completely rule out selection effects, for example related to early life mortality. However, confirmation of the F1–F2 results in the next generation F2–F3 significantly strengthens the results and allowed us to cope with the potential selection effects for IPs. In addition, a sensitivity analysis for sample size, in which we fitted all our statistical models on half of the UPDB and half of the LINKS data, provided similar results to the full sample results, indicating that our results are robust for a reduced sample size (Supplementary Figure 9).

Human Lifespan (defined as age at death) has a low heritability in the population at large[1–4]. Studies estimated the heritability of lifespan between 12% and 25%[1–3] and a recent study estimated that the heritability of lifespan was even lower, ~7%, after adjustment for the lifespans of nongenetic (in-law) relatives[4]. Therefore lifespan-based gene mapping may not be fruitful. In addition, the genetic component of lifespan includes the heritability of early life mortality, which is mainly due to disease and external causes. Despite the low heritability and polygenic architecture[26,37] of lifespan, recent genetic studies have identified[31,32] and replicated[53] some lifespan loci of which the rare alleles lower the risk of age-related diseases. Hence, using the lifespan trait hampers the identification of genetic loci contributing to survival into extreme ages (longevity). Longevity however, clusters strongly within families as shown by previous studies[5–9] and robustly quantified in this study. Hence, the longevity trait is much more promising and appropriate for the identification of genetic loci contributing to survival into extreme ages and should not be confused with the lifespan trait[1,12]. Our results imply that to find loci that promote survival to the highest ages in the population, genetic studies should be based on long-lived cases including at least parental mortality information but preferably also mortality information of siblings and other first and second-degree relatives. The longevity threshold should include cases belonging up to the top 10% survivors, with parents belonging up to the top 15% survivors of their birth cohort and siblings belonging up to the top 10% survivors of their birth cohort. To sharpen the longevity effect, the percentile threshold applied may be made more extreme but would likely lead unnecessarily to a sample size with limited power. If our proposed longevity definition is consistently applied across studies, the comparative nature of longevity studies may improve and facilitate the discovery of novel genetic variants.

## Methods

**Ethical regulations.** We complied with all relevant ethical regulations. For the Utah data (UPDB), the study was approved by the Resource for Genetic and Epidemiologic Research (RGE). For the Zeeland (LINKS) data, the study was approved by the International Institute of Social History. For this study, no informed consent needed to be obtained.

**Utah Population Database.** The UPDB contains demographic and genealogical information which is linked to medical records. The data construction began in the mid-1970s with genealogy records from the archives at the Utah Family History Library and was initially based on the founding members of the Utah population, their descendants, and then subsequently all individuals living in Utah. These records contain demographic and mortality information on the pioneers of Utah (US), their parents and children, and have been linked into multigenerational pedigrees. The founding families were selected for the UPDB when at least one member had a vital event (birth, marriage, or death) on the Mormon pioneer trail or in Utah. The UPDB has been expanded to incorporate other high-quality, statewide data sources, such as birth and death certificates, cancer records, driver license records, and census records. Currently, the UPDB contains information on more than 11 million individuals and covers a maximum of 17 generations[54] (https://healthcare.utah.edu/huntsmancancerinstitute/research/updb/data/family-records.php).

**LINKing System for historical family reconstruction.** The LINKS data contains demographic and genealogical information which was derived from linked vital event registers (birth, marriage, and death certificates). The data indexing began in 1995 by the "Zeeuws" archive and the results were published by way of "Wie-WasWie". The data currently covers over 25 million Dutch vital event records[55] (https://socialhistory.org/en/hsn/linking-system-historical-family-reconstruction-links). Data construction has been completed for the province of Zeeland and is still ongoing for the other provinces in the Netherlands. Currently LINKS Zeeland (henceforth referred to as LINKS) contains 739,453 birth, 387,102 marriage, and 641,216 death certificates which were linked together to reconstruct intergenerational pedigrees and individual life courses[47]. In total, the Zeeland data contains 1,930,157 persons covering a maximum of 7 generations[56].

**Historical context of Utah and Zeeland.** Both Utah and Zeeland were high fertility populations[46,48,57], with a mean number of children of around 7 during the period of this study (1740–1952). In general, Utah was marked by healthy living conditions and Zeeland by contrast, was a much unhealthier place to live. One of the main reasons for the unhealthy living conditions in Zeeland was the lack of clean drinking water, the high prevalence of waterborne diseases and of malaria[46,58,59]. In Utah the quality of the drinking water was good, since water from melting snow, that was filtered running of the mountains, was used to drink[48]. The differences in living conditions between Utah and Zeeland were reflected by a relatively low infant and childhood mortality in Utah[60] and high mortality rates for infants and children in Zeeland[59], especially before 1900. Moreover, Utah was known to be a high in-migration population[49] whereas there were indications that Zeeland had a low influx and outflux of migrants[47].

**Study selection.** For the current study, we used 3 filial (F) generations (F1–F3) from the UPDB and LINKS ($N_{UPDB+LINKS} = 314,819$). We reconstructed families in both datasets and denote generation 1 as the starting point of the pedigrees in the data. The starting point for this study was generation 3 because starting here minimized missing family links and birth or death dates due to the nature of the source material underlying the data. We denote generation 3 as filial generation 1 (F1). Subsequently, the children ($N_{UPDB+LINKS} = 123,599$) of the F1 parents were identified (F2) so that unique families were represented by 2 parents (F1) and their offspring (F2). Next an IP (F2) was randomly selected per F2 sibship ($N_{UPDB+LINKS} = 20,360$) meeting the following criteria: (1) the date of birth and death had to be available, (2) at least one child, sibling, and spouse had to be available, (3) sex had to be available, (4) for the UPDB data only, the IP should preferably be identifiable on a genealogy record (Supplementary Table 12). From there we identified the siblings (F2, $N_{UPDB+LINKS} = 108,122$), spouses (F2, $N_{UPDB+LINKS} = 22,018$), and the children (F3, $N_{UPDB+LINKS} = 123,599$) of the IPs (Table 1 and Fig. 1). To summarize, both in the UPDB and LINKS we identified IPs (F2), their parents (F1), siblings (F2), spouses (F2), and children (F3).

All individuals in LINKS have at some point in their lives lived in Zeeland, this is because the data were constructed based on vital event records from Zeeland. Utah was first settled in 1847 and in the UPDB mortality information for ancestors of Utah associated persons are available. As a result not all persons necessarily had to live in Utah. Supplementary Table 13 shows that in our data, 97% of the IPs lived in Utah. This percentage is lower for their fathers (80%) and mothers (87%), and is an expected pattern given the historic nature of how Utah was settled. Furthermore, 70% of the siblings, 97% of the spouses, and 92% of the children lived in Utah. The majority of the persons from our sample who lived in Utah, migrated from another state in the US to Utah (87%), 12% came from Europe, and 1% from the rest of the world.

**Lifetables.** We used cohort lifetables to calculate birth cohort and sex-specific survival percentiles for each individual in the UPDB and LINKS. This approach prevents against the effects of secular mortality trends over the last centuries and

enables comparisons across study populations[1,12]. We could not use US lifetables because cohort lifetables were not available and period lifetables were only available from 1933 onward. Moreover, the US birth cohort-based central death rates were generally incomplete at the earlier cohorts (up to 1900) and proved to be of limited use for our analyses. However, for Sweden and the Netherlands, population-based cohort lifetables were available from 1751 and 1850 until 2018 respectively[61–64]. These lifetables contained, for each birth year and sex, an estimate of the hazard of dying between ages $x$ and $x + n$ ($h_x$) based on yearly intervals ($n = 1$) up to 99 years of age. Conditional cumulative hazards ($H_x$) and survival probabilities ($S_x$) were derived using these hazards. In turn, we could determine the sex and birth year specific survival percentile for each person in our study. Swedish cohort lifetables date back furthest of all available lifetables and were shown to be consistent with the lifetables of multiple industrialized societies[65]. In addition, we ensured that the survival percentiles were calculated in the same way for the UPDB and LINKS to make a fair comparison between the survival percentiles. Hence, the Swedish cohort lifetables were used for both datasets and for the LINKS data the Dutch lifetables were used as a sensitivity analysis. Supplementary Figure 10 shows the ages at death corresponding to the top 10%, 5%, and 1% survivors for the UPDB and LINKS. This figure can be used to map the percentiles, which are based on percentile–age pairings from the Swedish lifetables, to absolute ages. For example: a top 10% female in 1750 matched an age of 76 years whereas this was 74 years for males. In 1850, a top 10% female and male matched an age of 83 years and 81 years, respectively.

**Statistical analyses**. Statistical analyses were conducted using R version 3.4.1[66]. We reported 95% confidence intervals (CIs) and considered $p$-values statistically significant at the 5% level ($\alpha = 0.05$).

**IP survival at increasing survival percentiles of relatives**. Analysis 1: To determine if (1) the association between the survival (measured as age at death) of IPs and the survival percentiles of their parents and siblings increased with increasing survival percentiles, and (2) a larger level of family aggregation, in terms of numbers of parents and siblings, was more evident at extreme survival percentiles, we investigated the association between IP survival and the number of parents and siblings reaching increasingly more extreme survival percentiles. We sequentially identified the number of parents and siblings belonging to the top $x$ ($x = 1, 2, 3, …, 60$) percentiles of their birth cohorts (from here: percentiles) and we analyzed their association with the survival of the IPs for each subsequent survival percentile using a Cox proportional hazard model:

$$\lambda\left(t_{ij}\right) = \lambda_0\left(t_{ij}\right)\exp\left(\boldsymbol{\beta Z}_{ij} + \boldsymbol{\gamma X}_{ij}\right) \quad (1)$$

where $t_{ij}$ is the age at death for IP $j$ in family $i$. $\lambda_0(t_{ij})$ refers to the baseline hazard, which is left unspecified in a Cox-type model. $\boldsymbol{\beta}$ is the vector of regression coefficients for the main effects of interest ($\boldsymbol{Z}$) which correspond to (1) the number of parents belonging to the top $x$ percentile, (2) and the number of siblings belonging to the top $x$ percentile. $\boldsymbol{\gamma}$ is a vector of regression coefficients for the effects of covariates and possible confounders ($\boldsymbol{X}$) which are IPs' religion (UPDB only), sibship size, birth cohort, sex, socio-economic status, mother's age at birth, birth order, birth intervals, and twin birth.

**Identifying a survival threshold that demarcates longevity**. Analysis 2: The previous analysis, based on the cumulative effects, does not allow us to identify a specific threshold to define longevity, since the top $x$ percentiles were not mutually exclusive, i.e., if a person belonged to the top 1% survivors, this person also belonged to the groups of top 5% and top 10% survivors. To determine the survival percentile threshold that drove the cumulative top $x$ percentile effects described in the previous section, we grouped IPs according to the survival of their parents and siblings for two separate analysis. More specifically, we constructed mutually exclusive groups of IPs based on having at least one parent or sibling belonging to group g (g = 1, 2, 3, …, 6): group 1 = [≥0th & ≤1th percentile], group 2 = [≥1th & ≤5th percentile], group 3 = [≥5th & ≤10th percentile], group 4 = [≥10th & ≤15th percentile], group 5 = [≥15th & ≤20th percentile], group 6 = [≥20th & ≤100th percentile]. Group membership was defined by the most long-lived parent or sibling of the IP. Using Cox proportional hazards models (see expression (1)), we compared the effects of all groups to reference group 6, corresponding to IPs with all parents or siblings belonging to the 20th or less extreme survival percentile and multiple combinations of defining group 6 were tested. Here, the $\boldsymbol{\beta}$ is the vector of regression coefficients for the main effects of interest ($\boldsymbol{Z}$) which correspond to (1) the IPs who were divided into mutually exclusive groups by their parental mortality and (2) the IPs who were independently grouped by their sibling mortality. Other parts of the expression are the same as noted in expression (1).

**Top 10% relatives and covariates in an integrated design**. Analysis 3: Based on the analyses expressed in the previous section, we chose the top 10% survivors for specific follow-up analyses. Based on the results presented in the cumulative and mutually exclusive group analyses, we focused on the top 10% surviving family members because the mutually exclusive group analysis (analysis 2) indicated longevity effects for siblings beyond the top 10% and 15% for siblings and parents,

respectively. Using the top 10% is consistent between the two groups and is a conservative choice. Furthermore, the cumulative analysis (analysis 1) indicated that the top 10% was a good trade-off between effect size and group size (power) within and between the UPDB and LINKS. Hence, we focused on top 10% parents and siblings in an integrated design to investigate the association between IP survival and the number of parents and siblings belonging to the top 10%. We subsequently investigated the association between the number of top 10% siblings and IP survival for IPs without top 10% parents, using Cox regression (see expression (1)). Here the $\boldsymbol{\beta}$ is the vector of regression coefficients for the main effects of interest ($\boldsymbol{Z}$) which correspond to (1) the number of parents and siblings belonging to the top 10% and (2) the number of siblings belonging to the top 10% for IPs without top 10% parents. Other parts of the expression are the same as noted in expression (1).

In all Cox regression analyses, based on expression (1), we accounted for the fact that IPs were selected to have a spouse and at least one child (left truncation) by using an IP specific age at entry in the study based on the IP's age at first child or the age at marriage, whichever was later. A similar approach was followed for the spouses of the IPs and no adjustment for left truncation was necessary for the children of the IPs, since they were not selected in any way. Moreover, we accounted for right censoring in all relatives of the IPs. We furthermore adjusted for religion (UPDB only), sibship size, birth cohort, sex, socio-economic status, mother's age at birth, birth order, birth intervals, and twin birth since these are known to influence human survival[1]. Socio-economic status was constructed according to the Integrated Public Use Microdata Series (IPUMS) occupational coding scheme of 1950 (OCC1950)[67]. Importantly, for the sibling contribution to the cumulative percentile analysis (analysis 1), the sibling contribution to the top 10% analyses (analysis 3), and in all mutually exclusive group analyses (analysis 3), we used analytical weights when fitting the Cox models to avoid family size confounding. Adjustment was not necessary for the number of parents because this number is two by definition. However, sibship sizes vary. For example, a hypothetical IP with 4 siblings belonging to percentiles 1, 6, 8, and 30 will contribute with a weight $w = 3/4$ in the first analysis, based on the cumulative percentiles, when considering the top 10%. This same IP, when considering the top 5% will contribute with less weight, namely $w = 1/4$. In this way, each person contributed the same to the overall analysis across all percentiles. In the second analysis based on mutually exclusive groups, this same hypothetical IP would be assigned to g1, and will contribute to the analysis with a weight $w = 1/4$. In analysis 3, based on the top 10%, the IP will contribute with a weight of $w = 3/4$. In this way, we avoid a potential advantage of larger families to be represented in more extreme groups. Finally, we checked the proportional hazards and linearity assumptions in all fitted Cox models. We did not find evidence that model assumptions were violated for the main effects (parent/sibling and IP/children of IPs associations). The proportional hazards assumption was violated for some covariates. In such a case, stratification was applied for that covariate and this was mentioned in the legend of the table/figure.

**Verification of the results in a subsequent generation**. Analysis 4: To verify our results regarding the top 10% parents and siblings (analysis 3) in a subsequent generation (children, F3), we investigated whether children of top 10% IPs had a survival advantage compared to children of non-longevous IPs and whether this effect is stronger if the spouse of the IP also belonged to the top 10%. We further investigated familial clustering of longevity by studying the number of top 10% aunts and uncles of the children of IPs. A Cox-type random effect model was used:

$$\lambda\left(t_{ij}\right) = u_i\lambda_0\left(t_{ij}\right)\exp\left(\boldsymbol{\beta Z}_{ij} + \boldsymbol{\gamma X}_{ij}\right) \quad (2)$$

where $t_{ij}$ is the age at death or the age at last follow-up for child $j$ in family $i$, $\lambda_0(t_{ij})$ refers to the baseline hazard, which is left unspecified, $\boldsymbol{\beta}$ is a vector of regression coefficients for the main effects of interest ($\boldsymbol{Z}$) which correspond to (1) having a parent top 10% survivor in a first analysis and (2) the effect of the number of uncles/aunts (F2) top 10% in a second analysis. $u > 0$ refers to an unobserved random effect (frailty) shared by F3 children of a given IP. This unobserved heterogeneity shared within sibships was assumed to follow a log-normal distribution. $\boldsymbol{\gamma}$ contains the effect of person-specific covariates $\boldsymbol{X}$, similar to those included in the previous analyses.

**Survival of spouses by the longevity of the index persons**. Analysis 5: To investigate the survival of spouses, we applied a group approach, similar to that used above, and analyzed the groups with Cox regression. We grouped the spouses by the survival of the IPs creating 6 different groups g (g = 1, 2, 3, …, 6): group 1 = [≥0th & ≤ 1th percentile], group 2 = [≥1th & ≤ 5th percentile], group 3 = [≥5th & ≤10th percentile], group 4 = [≥10th & ≤15th percentile], group 5 = [≥15th & ≤ 20th percentile], group 6 = [≥20th & ≤100th percentile]. We compared the groups in two steps: (1) group 6 was the reference category and (2) comparing all groups with each other (post-hoc), applying a Bonferroni correction for multiple testing:

$$\lambda\left(t_{ij}\right) = \lambda_0\left(t_{ij}\right)\exp\left(\boldsymbol{\beta Z}_{ij}\right) \quad (3)$$

where $t_{ij}$ is the age at death or the age at last follow-up for spouse $j$ in family $i$.

$\lambda_0(t_{ij})$ refers to the baseline hazard, which is left unspecified in a Cox-type model. $\beta$ is the regression coefficient referring to the main effects of interest ($Z$), which are the spouses who were divided into mutually exclusive groups by the IPs mortality.

**Code availability**. The scripts containing the code for data pre-processing and data analyses can be freely downloaded at: https://git.lumc.nl/molepi/PUBLIC/Longevity_top10perc_survivors. This repository describes the main analyses done.

## Data availability

The UPDB and LINKS data that support the findings of this study are available from the UPDB and the IISG but restrictions apply to the availability of these data, which were used under license for the current study, and so are not publicly available. Data are however available upon reasonable request and with approvals of the UPDB and the IISG. The LINKS data is available upon request to Dr. Kees Mandemakers (kma@iisg.nl). The UPDB data is also available upon request and approval by the Resource for Genetic and Epidemiologic Research (RGE). More information on making this request can be obtained from one of the authors, Dr. Ken R. Smith (ken.smith@fcs.utah.edu).

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

## Acknowledgements

This research was supported by Netherlands Organization for Scientific Research [360-53-180], the Professor van Winterfonds, and the Leiden University Fund [6223/07-06-2016]. Niels van den Berg thanks Professor Ken R. Smith for hosting his stay in Utah (US) to work with the Utah Population Database. The authors also thank Dr. Joris Deelen and Dr. Thies Gehrmann for providing their input to this work as independent readers.

## Author contributions

Niels van den Berg is the study investigator and was responsible for initiating the study, data management, data analyses, writing the first draft of the manuscript and finalizing it, and obtaining funding to visit Ken R. Smith in Utah. Angelique Janssens and P. Eline Slagboom are the study principal investigators who conceived and obtained funding for the project, which this study is a part of. Ken R. Smith is the head of the UPDB and he hosted the stay of Niels van den Berg in Utah, provided access to the UPDB, and supervised all that concerned the UPDB within this study. P. Eline Slagboom and Marian Beekman provided overall project coordination and supervision. Mar Rodriguez-Girondo provided overall statistical analyses coordination and supervision and assisted in the overall project coordination. Kees Mandemakers is the head of LINKS and provided access and support to the LINKS data. Rick Mourits provided access to documentation and conversion tables of socio-economic status coding schemes between the two databases. Ingrid van Dijk assisted with the initial family reconstruction process and assisted in working with the LINKS data.

## Additional information

**Competing interests:** The authors declare no competing interests.

