## [Peer Review File · Nature Communications]

Reviewer #1 (Remarks to the Author):

This is an interesting paper, although some of these findings are not new. The paper has several problems, which need to be resolved.

The major concern of this paper is the use of truncated records in the survival analyses. The authors wrote: "No IPs were censored, as they were selected to have an available birth and death date." This would be acceptable if they use extinct birth cohorts with members who already died out. However they use IPs born between 1767 and 1929. It is clear that many persons born after 1902 are still alive and their lifespan is unknown. If authors do not use censored observations then the sample would have truncated records, which would distort the results. The same critique applies for sibling and children data.

There are also some other issues:

1. If the authors attempt to determine the threshold for longevity, it would be good to discuss earlier attempts of finding longevity threshold. For example, it was found that heritability of lifespan is not linear and is higher in the case of long-lived parents. Later this nonlinearity in lifespan heritability was found in the study of twins. This phenomenon was used to establish a longevity threshold, which was estimated as a changing point of the dependence of person's lifespan on parental lifespan (Gavrilova, Gavrilov, When does human longevity start?: Demarcation of the boundaries for human longevity. *Journal of Anti-Aging Medicine*, 2001, 4(2): 115-124). This study was done on genealogies of European aristocratic families who did not experience serious environmental hardship. In the proposed paper the dependence between IP group (parental survival) and HR is more or less linear without obvious threshold. The authors ignore earlier studies of longevity threshold.
2. The analyses are not made separately for male and female IPs, so it is not clear whether these findings are the same or not for men and women.
3. Authors use limited number of environmental variables in their study and lament the absence of information about early-life conditions. At the same time it was already shown that a large set of childhood and middle-age variables do not modulate effects of paternal and maternal longevity on survival to age 100 years (Predictors of Exceptional Longevity: Effects of Early-Life and Midlife Conditions, and Familial Longevity. *North American Actuarial Journal*, 2015, 19:3, 174-186). These results are not cited in this paper.
4. Authors claim that the Human Mortality Database does not have data for cohort mortality in the U.S. In fact it has a set of age-specific central death rates for U.S. birth cohorts, which would allow authors to reconstruct the number of survivors (although not for all cohorts used in the study).

Smaller problems:

It is not clear why there is no information about mean values for N in the case of UPDB (Supplementary Table 2).

Supplementary figures 2-5. It looks like these figures have "age in years" on the x-coordinate. Instead, the authors use "survival in years," which is confusing taking into account that they worked with percentiles before.

The paper needs some editorial work. See, for example, fragment of sentence in the second paragraph of Discussion section.

Reviewer #2 (Remarks to the Author):

The work is interesting and novel, but not tremendously so. It adds some depth to what I think is a consensus that there is a genetic component to variation in human lifespan. The quantitative results have not been developed in such a way that would make them readily comparable to other studies of the same question. Overall, not enough effort has been made to explain how this paper fits in with the existing literature on inheritance of longevity on a quantitative level. The “threshold” of 10% seems like a tradeoff appropriate to this particular dataset — based on its size and genetic composition — rather than having any more general relevance.

An important strength of this study is the synoptic treatment of two very different data sets. The careful consideration of the cutoff between high and low survival is interesting, and a definite positive feature, compared to the common choice of an arbitrary cutoff. The statistical analysis is reasonably straightforward, and could be reproduced if one had access to the data. (I don't know how readily accessible the data are.) It would be helpful if the authors would make their data-cleaning and analysis scripts available as a git depository, or in some other form. The paper describes the statistical model and methodology well. There does not seem to have been any attempt made to verify model assumptions or test robustness. But this seems, on general principles, unlikely to be a major problem.

Ample consideration has been given to potential confounders, though it is not entirely clear about how adjustment was carried out. It's also not entirely clear whether the factors being adjusted for are all necessary, and they probably are not sufficient to confirm a genetic transmission of longevity. Some consideration of the implicit causal model would have been helpful.

There is a typo in `N_LINKS = 202,343` on line 98.

Reviewer #3 (Remarks to the Author):

This submission is an excellent paper with results of wide interest to demographers. It probes two rich historical datasets with families linked across three generations spanning two centuries and brings the data to bear on heritable, potentially genetic, influences on longevity. The analysis is thoroughly documented and wide-ranging. I found that questions that came to mind while reading the paper were frequently addressed later in the text. I recommend this paper for publication.

Here are some points where further discussion or revision might be helpful for readers:

- 1) Is the interval in the Cox model being measured from age zero, out to death or censoring, or from some other starting point? Since each Index Person (IP) has to have a child in order to enter the sample, presence in the sample is conditional on survival to some varying age. A word about how left-truncation is being handled would be helpful.

2) From the point of view of genetic demography, survival in the face of senescent mortality is of primary interest. An alternative to the authors' analysis would have been to restrict the sample to IPs over, say, 45, and start the Cox analysis at age 45. For these populations, maternal mortality would seem to be a complicating factor that might be good to avoid. This is not a request for the authors to do a different Cox fitting but instead to explain in more detail the reasons behind their choice.

3) The cohort-by-cohort levels of infant mortality in the Swedish lifetables can hardly be expected to capture those in the population in the Utah database, nor those in Zeeland over the centuries. An alternative would have been to calculate percentiles based on lifetables conditional on survival to, say, age 10 or 15. Again, an explanation of the authors' choice would be valuable.

4) The translation between percentiles and ages cohort-by-cohort in Supplementary Figure 1 is quite informative. A few examples of pairings of percentile and age from this figure might be mentioned in the main text. These are apparently the percentile-age pairings from the Swedish lifetables used to calculate percentiles in both datasets, Please say so. Is there some way to give the reader a sense of what the empirical percentiles for the IPs in the two datasets look like, subject to conditioning on survival to some fixed age?

5) The labeling on the Y-axis for the right-hand panels in Figure 2 is misleading. The words 'cumulative hazard' should appear in the label, not just in the caption. These panels would be more informative if the Y-axis only reached out to 1.5 or 2.0 or so and if the X-axis were cut off at 90 or so. In the present version, the range is too broad and the resolution too small for the curves to be easily distinguished by eye.

6) More discussion of the geographical spread of the IPs and their parents and children in the 'Utah' database would be welcome. The database was originally built around Utah founder families. But is it not true that many or most of the individuals who appear in the database were not living their lives in Utah?

7) In the Supplementary Information, a

histogram of the empirical values of the percentile points for the deaths of the IPs would be welcome, so that readers could compare them to a uniform distribution.

8) In the Supplementary Information, it would also be interesting to see a scatterplot of the percentile for the longer-lived parent versus the percentile for the IP.

Another option would be a scatterplot of the average of the father's and mother's percentiles versus the percentile for the IP, perhaps subsampled so that the scatter of points is more easily visible. Such plots would make it easier to visualize the (piecewise) parent-child correlations that underlie the Cox Model results, in a form analogous to Galton's famous plot for heights.

Reviewer: Kenneth W. Wachter, University of California, Berkeley

Reviewer 1

This is an interesting paper, although some of these findings are not new. The paper has several problems, which need to be resolved.

Comment A

The major concern of this paper is the use of truncated records in the survival analyses. The authors wrote: "No IPs were censored, as they were selected to have an available birth and death date." This would be acceptable if they use extinct birth cohorts with members who already died out. However they use IPs born between 1767 and 1929. It is clear that many persons born after 1902 are still alive and their lifespan is unknown. If authors do not use censored observations then the sample would have truncated records, which would distort the results. The same critique applies for sibling and children data.

Response

We thank the reviewer for pointing this out. The comment made us think again about the study design of this paper. The reviewer makes two important remarks about truncated lifespans; 1. Regarding the index persons (IPs) and 2. Regarding the siblings and children of the IPs. We will address the two remarks separately:

Siblings and children of IPs:

There is no direct selection on the siblings and children of IPs and thus we do allow, and take into account right censoring for all groups that can be censored (all groups except the IPs). This was described in the statistics section, paragraph 6 of the methods, but to clarify this better in the text, we have adjusted the methods so that it is now more explicitly stated.

IPs:

We would like to thank the reviewer for noticing this issue. We agree that the use of truncated records may induce systematic bias in subsequent survival analysis in the setting of this study (in the form of underestimated Hazard Ratios). Hence, we follow the advice of the reviewer and exclude IPs, and the relatives of these IPs, born after 1902. The results and subsequent conclusions were not affected by the excluded IPs and their relatives, though some effects (for example those illustrated in figure 3) are now marginally stronger which is in line with the potential of underestimated Hazard Ratios.

Implementations

We excluded 3 LINKS IPs and 683 UPDB IPs (and the families related to these IPs) who were part of potentially non-extinct birth cohorts (the birth cohorts after 1902) and adjusted all figures, tables and results in the text conform this adjustment. We would like to refer to the revised paper for the implementations.

We have also adjusted the methods so that it is more clear that right censoring is allowed and taken into account for all groups other than the IPs.

Chapter 4: Methods

paragraph 6 (line 544-549)

In all Cox regression analyses, based on expression 1, we accounted for the fact that IPs were selected to have a spouse and at least one child (left truncation) by using an IP specific age at

entry in the study based on the IP's age at first child or the age at marriage, whichever was later. A similar approach was followed for the spouses of the IPs and no adjustment for left truncation was necessary for the children of the IPs, since they were not selected in any way. Moreover, we accounted for right censoring in all relatives of the IPs.

Comment B

If the authors attempt to determine the threshold for longevity, it would be good to discuss earlier attempts of finding longevity threshold. For example, it was found that heritability of lifespan is not linear and is higher in the case of long-lived parents. Later this nonlinearity in lifespan heritability was found in the study of twins. This phenomenon was used to establish a longevity threshold, which was estimated as a changing point of the dependence of person's lifespan on parental lifespan (Gavrilova, Gavrilov, When does human longevity start?: Demarcation of the boundaries for human longevity. *Journal of Anti-Aging Medicine*, 2001, 4(2): 115-124). This study was done on genealogies of European aristocratic families who did not experience serious environmental hardship. In the proposed paper the dependence between IP group (parental survival) and HR is more or less linear without obvious threshold. The authors ignore earlier studies of longevity threshold.

Response

We agree with the reviewer that we did not sufficiently embed our findings in the already existing literature of longevity thresholds.

This comment also made us realize that we have to clarify our results regarding the mutually inclusive analysis (methods, analysis 1) and the mutually exclusive analysis (methods, analysis 2). The mutually inclusive analysis are meant to show the decrease in HR for IPs with each consecutive survival percentile increase of their parents and siblings and indicates that indeed the association between study persons and their relatives becomes increasingly stronger when those relatives lived longer. This is in line with earlier research from Sebastiani et al.¹. This mutually inclusive analysis is however not suitable to identify a specific threshold to define longevity, since the percentiles were not mutually exclusive, i.e., if a person belonged to the top 1% survivors, this person also belonged to the groups of top 5% and top 10% survivors. To investigate if the linear trend from figure 2 actually starts from a specific percentile we conduct analyses in mutually exclusive (methods, analysis 2) IP groups, meaning that the IP could only be in one group. From this we identify a threshold, based on parental and sibling longevity, at the top 10% survivors. Our findings are thus in line with Gavrilova and Gavrilov² regarding the existence of a turning point from whereon we can identify longevity family associations using phenotypic mortality indicators.

Implementations

We improved the embedding of our results in the longevity literature by adding two paragraphs in the discussion section.

Chapter 3: Discussion

paragraph 2 and 3 (line 276-297)

Previous studies of smaller sample size than the current study, usually focusing on two generations of selected data (for mortality or geographical locations), were the first (1) to indicate an increase in the heritability of lifespan with parental age^{3,4} and to show high recurrences risks between parental and offspring or sibling longevity. Thus, providing indications that the heritability of longevity may be stronger than that of lifespan⁵⁻⁹, (2) to indicate that sibling relative risks beyond the top 5% survivors might not increase in a linear fashion¹ and that this non-linearity may indicate the existence of a longevity threshold², and

(3) to identify longevity recurrence risks for siblings or parents of selected longevous individuals^{1,5,8,10-13} and show increased survival probabilities and longevity recurrence risks for children of longevous parents^{6-8,14-16}.

Here we use two unique, large three-generational datasets (314,819 individuals in 20,360 families) which were unselected for survival and cover multiple geographical areas. We utilized these datasets to robustly identify a longevity threshold by showing that the association between IP survival and the survival of parents and siblings was not linear but in fact was driven by the oldest, up to the top 10%, surviving parents and siblings. We further showed that the survival of F2 IPs (and their F3 children) increased with each additional longevous parent (F1 and F2), and sibling (F2). We extended these analyses by showing that the survival for children of IPs increased with each additional longevous aunt or uncle. We also extended the analyses by investigating the association between IP survival and sibling longevity, and between the survival of IPs' children and the longevity of their aunts or uncles in the absence of longevous parents.

Comment C

The analyses are not made separately for male and female IPs, so it is not clear whether these findings are the same or not for men and women.

Response

We appreciate the reviewers interest in the sex specific analyses for IPs and we agree with the reviewer that the results may of interest of the readers. We indeed only provided the sex specific parental analyses in supplemental table 3 and 9. We now added the results for the sex specific IP analyses in supplementary figure 7 and supplementary table 10. Similar to the results from supplementary table 3 and 9, no differential association was identified between the number of longevous parents/siblings and male or female IPs or the children of IPs.

Implementations

We added a supplementary figure (supplementary figure 7) showing a sex specific version of figure 2. We furthermore added a supplementary table (supplementary table 10) showing a sex specific version of table 2 and 3, using interaction terms between the number of longevous parents and sex. We added these results to the results and discussion sections of the paper.

Chapter 2: Results *paragraph 8 (line 215-220)*

In a final step, we observe no evidence that the association of IP survival and parental longevity depends on maternal or paternal effects, for example through transmission preferentially via the mother or father (Supplementary Table 3). Likewise, the association of IP survival and parental longevity does not depend on the sex of the IPs, meaning that this association is equal for sons and daughters (Supplementary Figure 7 and Supplementary table 10).

Chapter 2: Results
 paragraph 9 (233-236)

Similar to the IPs, we observe: (1) that the survival of children does not depend on maternal or paternal effects (Supplementary Table 3) and (2) that the association between parents and offspring is equal for sons and daughters (Supplementary table 10).

Chapter 2: Results
 Supplementary figure 7

- Green lines present 0 top x percentile parents, yellow lines represent 1 top x percentile parents, blue lines represent 2 or 2+ longevous parents.
- Round nodes: males, triangle nodes: females.

Chapter 2:Results
 Supplementary table 10

Supplementary table 10: Sex specific survival analysis for IP's by top 10% parents

		UPDB			LINKS		
		N (mean)	HR (95% CI)	p-value	N (mean)	HR (95% CI)	p-value
F1-F2 (IP)	Top 10% parents (F1)						
	0	6640 (0.65)			7861 (0.78)		
	1	3167 (0.31)	0.85 (0.75-0.96)	0.0067	2096 (0.20)	0.83 (0.78-0.91)	<0.0001
	2	439 (0.4)	0.68 (0.59-0.80)	<0.0001	184 (0.2)	0.61 (0.48-0.76)	<0.0001
	Sex						
	Man (ref)	5053 (0.49)			4776 (0.48)		
	Women	5193 (0.51)	0.73 (0.68-0.78)	<0.0001	5338 (0.52)	1.02 (0.97-1.08)	0.3600
	Sex* Top 10% parents (F1)						
	0 Top 10 parents women (ref)	6640 (0.65)			7861 (0.78)		
	1 Top 10 parents women	3167 (0.31)	0.92 (0.78-1.10)	0.3440	2096 (0.20)	0.96 (0.86-1.10)	0.4041
2 Top 10 parents women	439 (0.4)	1.10 (0.89-1.35)	0.4010	184 (0.2)	1.36 (0.97-1.91)	0.0748	
F2-F3 (Child of IP - one LL parent)	Top 10% parents (F1)						
	0	48619 (0.80)			53378 (0.85)		
	1	12179 (0.20)	0.83 (0.77-0.89)	<0.0001	9096 (0.15)	0.86 (0.78-0.94)	<0.0017
	Sex						
	Man (ref)	31258 (0.51)			32136 (0.52)		
	Women	29540 (0.49)	0.56 (0.55-0.58)	<0.0001	30338 (0.48)	0.64 (0.63-0.66)	<0.0001
	Sex* Top 10% parents (F1)						
	0 Top 10 parents women (ref)	48619 (0.80)			53378 (0.85)		
	1 Top 10 parents women	12179 (0.20)	1.03 (0.98-1.08)	0.2390	9096 (0.15)	1.00 (0.94-1.05)	0.1863
	F2-F3 (Child of IP - two LL parents)	Top 10% parents (F1)					
0		58672 (0.96)			60962 (0.97)		
2		2126 (0.04)	0.68 (0.57-0.80)	<0.0001	1512 (0.03)	0.78 (0.63-0.96)	0.0138
Sex							
Man (ref)		31258 (0.51)			32136 (0.52)		
Women		29540 (0.49)	0.56 (0.55-0.58)	<0.0001	30338 (0.48)	0.64 (0.63-0.66)	<0.0001
Sex* Top 10% parents (F1)							
0 Top 10 parents women (ref)		58672 (0.96)			60962 (0.97)		
2 Top 10 parents women		2126 (0.04)	1.09 (0.99-1.21)	0.0812	1512 (0.03)	1.00 (0.87-1.13)	0.1947

-Means represent a mean for a continuous variable and a proportion for a categorical variable

-Additional covariates are: religion, sibship size, birth cohort, sex, socio-economic status, mother's age at birth, birth order, birth intervals, and twin birth.

-PA=father, MA=mother, NL=non-longevous, LL=longevous, NL=non-longevous, longevous was defined as belonging to the top 10% of a persons' birth cohort.

-Here P-values were rounded to 4 digits.

Comment D

Authors use limited number of environmental variables in their study and lament the absence of information about early-life conditions. At the same time it was already shown that a large set of childhood and middle-age variables do not modulate effects of paternal and maternal longevity on survival to age 100 years (Predictors of Exceptional Longevity: Effects of Early-Life and Midlife Conditions, and Familial Longevity. North American Actuarial Journal, 2015, 19:3, 174-186). These results are not cited in this paper.

Response

We thank the reviewer for pointing out this paper, which we overlooked. We unfortunately cannot cover all environmental influences with our genealogical data. We do cover childhood mortality in F3 but indeed cannot cover this in the F2 IPs. The covariates we included are known to associate with mortality and we included them to, as good as possible, adjust for environmental influences. It is interesting that the “Predictors of Exceptional Longevity” paper identifies results similar to our analyses. We incorporated the paper in our discussion.

Implementations

We included a section in the discussion where we discuss the association between the environmental covariates and the parental/sibling longevity indicators in the context of the current literature.

Chapter 3: Discussion

paragraph 7 (line 351-360)

Nevertheless, these covariates neither confounded the association between parental (F1) and sibling (F2) longevity and IP (F2) survival, nor that between IP (F2) and spouse (F2) longevity and their children’s (F3) survival or between longevity of aunts and uncles (F2) and the survival of IPs’ children (F3). This is in line with previous studies showing only a minor¹⁷ or no¹⁸ influence of environmental covariates on the association between parental longevity and offspring survival. It was also shown in that a range of early life factors, such as farm ownership, parental literacy, and parental occupation did not affect the association between parental and offspring mortality¹⁸. We, however, cannot completely rule out that other, unobserved non-genetic familial effects may affect our results.

Comment E

Authors claim that the Human Mortality Database does not have data for cohort mortality in the U.S. In fact it has a set of age-specific central death rates for U.S. birth cohorts, which would allow authors to reconstruct the number of survivors (although not for all cohorts used in the study).

Response

We appreciate this remark of the reviewer about the availability of cohort and age-specific US death rates, which range from 1852 - 1986, in the Human Mortality Database. Unfortunately the death rates are not usable for our research because childhood, early life, and for the earliest birth cohorts also late life mortality information is unavailable. In the first birth cohort of 1852, the death rates start at age 81 and slowly more information becomes available over the subsequent birth cohorts. The mortality information is however to

inconsistently and incompletely available for us to do a sensitivity analysis with the US death rates. Though the US death rates could not be used for our study, we acknowledge that our claim was too bold.

Implementations

We have adjusted our claim about the absence of cohort mortality information in the United States.

Chapter 4: Methods

paragraph 6 (line 459-464)

We could not use United States (US) lifetables because cohort lifetables were not available and period lifetables were only available from 1933 onward. Moreover, the US birth cohort based central death rates were generally incomplete at the earlier cohorts (up to 1900) and proved to be of limited use for our analyses. However, for Sweden and the Netherlands, population based cohort lifetables were available from 1751 and 1850 until 2018 respectively¹⁹⁻²².

Comment F1

It is not clear why there is no information about mean values for N in the case of UPDB (Supplementary Table 2).

Response

We thank the reviewer for pointing out this mistake. This probably slipped through our final checks of the figures and tables.

Implementations

We added the mean values and did an additional check of all figures and tables for inconsistencies.

Chapter 2: Results

Supplementary table 2

Supplementary table 2: Survival analysis for IP's by top 10% siblings among IPs without top 10% parents

UPDB			
non-longevous parents			
	N (mean)	HR (CI)	P-value
Top 10% sibs of RP			
0 (ref.)	4639 (0.70)		
1	1473 (0.22)	0.85 (0.79-0.91)	<0.0001
2+	528 (0.8)	0.78 (0.67-0.90)	<0.0001
LINKS			
non-longevous parents			
	N (mean)	HR (CI)	P-value
Top 10% sibs of RP			

0 (ref.)	6867 (0.87)		
1	886 (0.11)	0.78 (0.72-0.85)	<0.0001
2+	108 (0.2)	0.72 (0.53-0.99)	0.0429

-Table corresponds to the CH curves in the top and bottom right panel of figure 2

-Means represent a mean for a continuous variable and a proportion for a categorical variable

-Additional covariates are: religion, sibship size, birth cohort, sex, socio-economic status, mother's age at birth, birth order, birth intervals, and being a twin

-Here P-values were rounded to 4 digits

Comment F2

Supplementary figures 2-5. It looks like these figures have “age in years” on the x-coordinate. Instead, the authors use “survival in years,” which is confusing taking into account that they worked with percentiles before.

Response

We agree with the reviewer that using the term survival on the x-axis can be confusing. In order to emphasize that we modeled follow up time for the IPs using age at death in years as time scale, we changed the labeling of the x-axis to “follow up time (age in years)”

This comment also made us realize that we should clarify in the introduction of the paper that we measure survival (follow up time) for the IPs in terms of age at death and for the children of IPs in terms of age at death or age at last observation. It was now only explained in the methods how survival was measured for IPs and children of IPs (methods, paragraph 3 (line 454-461) and paragraph 8 (line 535-542) of the statistical analysis section).

Implementations

We changed the labeling of the x-axis to in figure 2E-H and supplementary figures 2-5 to “follow up time (age in years). We refer to the figures themselves for the adjusted x-axes labeling.

We further introduced the different terminology regarding survival percentiles (which were used as predictors in the cox models) and survival (the outcome the cox models), in terms of follow up time (age in years) in the introduction of the paper.

Chapter 1: Introduction paragraph 3 (line 78-95)

We use the data available in the Utah Population Database (UPDB,Utah) and the LINKing System for historical family reconstruction (LINKS,Zeeland) based on US and Dutch citizens, respectively. Zeeland was a region with difficult living conditions compared to Utah (see methods section). In these datasets we identify 20,360 three-generational families (F1-F3) containing index persons (IPs, F2), their parents (F1), siblings (F2), spouses (F2), and children (F3) comprising 314,819 persons in total. First, we examine the association between the survival, measured as age at death, of IPs (F2) and the number of parents (F1) and siblings (F2) belonging to the top 1-60% of their birth cohort, in a cumulative way (comparing mutually inclusive percentile groups). Second, we determine the survival percentile threshold that drives the cumulative effects as a criterion for defining human longevity by investigating IP (F2) survival when divided into mutually exclusive groups based on the longevity of their

parents (F1) and siblings (F2). Third, we focus on the top 10% parents and siblings to investigate whether longevous and non-longevous parents, with increasing numbers of longevous siblings, transmit longevity to the IPs. Fourth, we confirm our findings in the next generation (F3) by examining the association between the survival, measured as age at death or last observation, of IPs' children (F3) and longevity of IPs (F2), their spouses (parents, F2) and siblings (aunts and uncles, F2). Finally, we explore potential environmental influences by studying spouses (F2) of longevous IPs (F2).

Comment F3

The paper needs some editorial work. See, for example, fragment of sentence in the second paragraph of Discussion section.

Response

We agree with the reviewer that some errors slipped through our editorial checks.

Implementations

We have reviewed the entire paper, figures, and tables in order limit the number of editorial mistakes. We refer to the new manuscript for all adjustments.

Reviewer 2

The work is interesting and novel, but not tremendously so. It adds some depth to what I think is a consensus that there is a genetic component to variation in human lifespan. An important strength of this study is the synoptic treatment of two very different data sets. The careful consideration of the cutoff between high and low survival is interesting, and a definite positive feature, compared to the common choice of an arbitrary cutoff.

Comment A

The quantitative results have not been developed in such a way that would make them readily comparable to other studies of the same question.

Response

We thank the reviewer for this remark and would like to discuss and elaborate our considerations regarding the study design and analysis techniques. Studies investigating the familial clustering of lifespan and longevity used many different study designs and statistical techniques, such as a twin, pedigree or case-control design and linear regression, logistic regression or survival analysis, as reviewed in²³. These studies often focused on 2 generational data covering a specific geographical region or group of persons (such as centenarians). We use two large three-generational datasets which are unselected for survival and cover multiple geographical areas, allowing us to conduct our analyses almost free of selections. Our data also allows us to address multiple research questions and can be used for future follow-up research. In addition, our data contains a combination of non-censored and censored observations, resulting in our choice to use survival analysis. Moreover, advancing insights into genealogical data analysis for longevity research influenced the methodological decisions, regarding for example the use of lifetables (percentiles) to standardize analyses over birth cohorts and study populations^{1,23}. Nevertheless, we think that our study results can be compared to other studies of the same question. We further elaborate on this in comment B where we discuss the embedding in the longevity literature.

Implementations

No adjustments have been made

Comment B

Overall, not enough effort has been made to explain how this paper fits in with the existing literature on inheritance of longevity on a quantitative level.

Response

This remark overlaps with comment B of reviewer 1, where we answered the following: We agree with the reviewer that our findings may be better embedded in the already existing literature of longevity thresholds.

Implementations

We have added two sections to the discussion in which we put the results of our study in perspective of the literature on the inheritance of longevity and longevity thresholds. In addition we have embedded the other sections of the discussion better into the literature.

Chapter 3: Discussion

paragraph 2 and 3 (line 276-297)

Previous studies of smaller sample size than the current study, usually focusing on two generations of selected data (for mortality or geographical locations), were the first (1) to indicate an increase in the heritability of lifespan with parental age^{3,4} and to show high recurrences risks between parental and offspring or sibling longevity. Thus, providing indications that the heritability of longevity may be stronger than that of lifespan⁵⁻⁹, (2) to indicate that sibling relative risks beyond the top 5% survivors might not increase in a linear fashion¹ and that this non-linearity may indicate the existence of a longevity threshold², and (3) to identify longevity recurrence risks for siblings or parents of selected longevous individuals^{1,5,8,10-13} and show increased survival probabilities and longevity recurrence risks for children of longevous parents^{6-8,14-16}.

Here we use two unique, large three-generational datasets (314,819 individuals in 20,360 families) which were unselected for survival and cover multiple geographical areas. We utilized these datasets to robustly identify a longevity threshold by showing that the association between IP survival and the survival of parents and siblings was not linear but in fact was driven by the oldest, up to the top 10%, surviving parents and siblings. We further showed that the survival of F2 IPs (and their F3 children) increased with each additional longevous parent (F1 and F2), and sibling (F2). We extended these analyses by showing that the survival for children of IPs increased with each additional longevous aunt or uncle. We also extended the analyses by investigating the association between IP survival and sibling longevity, and between the survival of IPs' children and the longevity of their aunts or uncles in the absence of longevous parents.

Chapter 3: Discussion

paragraph 7 (line 351-360)

Nevertheless, these covariates neither confounded the association between parental (F1) and sibling (F2) longevity and IP (F2) survival, nor that between IP (F2) and spouse (F2) longevity and their children's (F3) survival or between longevity of aunts and uncles (F2) and the survival of IPs' children (F3). This is in line with previous studies showing only a minor¹⁷ or no¹⁸ influence of environmental covariates on the association between parental longevity and offspring survival. It was also shown in that a range of early life factors, such as farm ownership, parental literacy, and parental occupation did not affect the association between parental and offspring mortality¹⁸. We, however, cannot completely rule out that other, unobserved non-genetic familial effects may affect our results.

Comment C

The "threshold" of 10% seems like a tradeoff appropriate to this particular dataset — based on its size and genetic composition — rather than having any more general relevance.

Response

We understand the reviewer's concern regarding the generalization of our results. We would like to argue that the threshold of 10% can be generally applied for the following reasons:

1. LINKS and the UPDB are genetically diverse populations and persons in the two databases are unrelated to each other. Moreover, environmental circumstances differed significantly between the two regions covered in the databases (see methods section for detailed description). The fact that we find the same longevity threshold and generally the same HRs may indicate that the influence of genetic diversity in EU or US populations is likely to be small. shows that our findings are unlikely to be subject to genetic diversity. The large longevity genetic (GWA) studies include EU UK, US and Asian populations. The threshold advocated in our paper may be directly applicable to these studies but may need additional exploration studying in Asian populations.
2. Sample size plays a role in identifying statistically significant effects (i.e. it will affect statistical power) but plays a less important role on the estimated effect sizes (HRs). The choice of our cut-off point for longevity has been mainly driven by the estimated effect sizes using the largest sample size currently available. Hence we expect our results to be robust with respect to further increases of the sample size; it is unlikely that more data would influence the determination of the longevity threshold towards lower (for example top 15 or 20%) survival percentiles. We would like to note that increasing the sample size may increase the contrasts in the more extreme survival percentiles (represented by group 1-3) but these contrasts are already observed in our study.

Even though in general large samples are preferred over small samples, we would like to provide the reviewer with a robustness check for the threshold detection, using a smaller sample size. Especially, because potential replication studies will likely be smaller than this study. Hence, we randomly excluded half of our sample and re-did the analyses. The results of the sensitivity analyses led to the same conclusions as the original ones, indicating that our approach to identify a cut-off for longevity seems robust to changes in sample size. We include the results of this sensitivity analysis as supplementary figure 8 in the paper.

Implementations

We include the results of this sensitivity analysis as supplementary figure 8 in the paper and reference to it in the discussion and the methods section.

Chapter 3: Discussion

paragraph 7 (line 362-369)

As a result, the number of longevous persons was quite low in LINKS relative to the UPDB. Although the IPs were randomly selected, we could not completely rule out selection effects, for example related to early life mortality. However, confirmation of the F1-F2 results in the next generation F2-F3 significantly strengthens the results and allowed us to cope with the potential selection effects for IPs. In addition, a sensitivity analysis for sample size, in which we fitted all our statistical models on half of the UPDB and half of the LINKS data, provided similar results to the full sample results, indicating that our results are robust for a reduced sample size (Supplementary Figure 8).

-Figure illustrates results similar to main figure 2 and 3 with the number of IPs cut in half.
 -Top row: green lines present 0 top x percentile parents or siblings, yellow lines represent 1 top x percentile parents or siblings, blue lines represent 2 or 2+ longevous parents or siblings.
 Bottom row: green lines represent the reference category, which is group 6.

Comment D

The statistical analysis is reasonably straightforward, and could be reproduced if one had access to the data. (I don't know how readily accessible the data are.)

Response

We would like to direct the reviewer to the data availability statement in the methods section of the paper on line 594 to 599. The data is freely available to anyone who wants to use it but cannot be stored in a public repository due to legislative restrictions.

Implementations

No adjustments have been made

Chapter 4: Methods

Data availability statement

The UPDB and LINKS data that support the findings of this study are available from the UPDB and the IISG but restrictions apply to the availability of these data, which were used under license for the current study, and so are not publicly available. Data are however available upon reasonable request and with approvals of the UPDB and the IISG. The LINKS data is available upon request to Dr. Kees Mandemakers (kma@iisg.nl), director of the LINKS data at the International Institute of Social History, Cruquiusweg 31, 1019 AT Amsterdam, the

Netherlands. The UPDB is also available upon request and approval by the Resource for Genetic and Epidemiologic Research (RGE). More information on making this request can be obtained from one of the authors, Dr. Ken R Smith (ken.smith@fcs.utah.edu), director of the UPDB at the University of Utah, 225 S. 1400 E. Rm 228 Salt Lake City, United States.

Comment E

It would be helpful if the authors would make their data-cleaning and analysis scripts available as a git depository, or in some other form.

Response

We fully agree with the reviewer. We made our scripts publicly available in the following git repository: https://git.lumc.nl/molepi/PUBLIC/Longevity_top10perc_survivors

Implementations

We added the GitLab repository link and a small description to the data availability statement.

Chapter 4: Methods

Code availability statement

Code availability

The scripts containing the code for data pre-processing and data analyses can be freely downloaded at: https://git.lumc.nl/molepi/PUBLIC/Longevity_top10perc_survivors. This repository describes the main analyses done.

Comment F

The paper describes the statistical model and methodology well. There does not seem to have been any attempt made to verify model assumptions or test robustness. But this seems, on general principles, unlikely to be a major problem.

Response

We thank the reviewer for this relevant remark. We have tested all model assumptions regarding the Cox regressions. We did not find evidence for the violation of the proportional hazards assumption for the main effects (parental, sibling, and aunts and uncle longevity categories) in any of the analyses. This can be seen by visual inspection of the cumulative hazards in figure 2 and supplementary figures 2-6. In addition, we fitted statistical tests using the `cox.zph` package in R to formally test the proportional hazards assumptions. This test is designed to assess the null hypothesis that the hazards are proportional. Hence a P-value < 0.05 is an indication that the hazards are not proportional. None of the main effects had a P-value smaller than 0.05, except for the 2 parent effect for the top 10% analysis which was on the border. The combination of the visual inspection and the official testing indicated no significant violations of the proportional hazards assumption for the main effects.

For some of the covariates the proportional hazards assumption was significantly violated. If this was the case we stratified for this covariate and mentioned it in the legends of the

applicable figure or table. Furthermore, assumptions for linear dependence of continuous covariates with the hazards have also been investigated, and no violation was identified.

Implementations

We now explicitly mentioned in the methods section that no violations of proportional hazards were identified for the main effects and that covariates were stratified if the proportional hazards assumption was violated.

Chapter 4: Methods

paragraph 6 (line 563-592)

In the second analysis based on mutually exclusive groups, this same hypothetical IP would be assigned to g1, and will contribute to the analysis with a weight $w=1/4$. In analysis 3, based on the top 10%, the IP will contribute with a weight of $w3/4$. In this way we avoid a potential advantage of larger families to be represented in more extreme groups. Finally, we checked the proportional hazards and linearity assumptions in all fitted Cox models. We did not find evidence that model assumptions were violated for the main effects (parent/sibling and IP/children of IPs associations). The proportional hazards assumption was violated for some covariates. In such a case, stratification was applied for that covariate and this was mentioned in the legend of the table/figure.

Comment G

Ample consideration has been given to potential confounders, though it is not entirely clear about how adjustment was carried out. It's also not entirely clear whether the factors being adjusted for are all necessary, and they probably are not sufficient to confirm a genetic transmission of longevity. Some consideration of the implicit causal model would have been helpful.

Response

We appreciate the reviewer's comment that we gave ample consideration to potential covariates/confounders. The covariates we included are known to associate with mortality²³ and we included them to, as good as possible, adjust for environmental influences. We have fitted several models for all our analyses, with and without the covariates to investigate whether the parental and sibling effects were affected by the covariates and to eliminate environmental variance. We agree that adjustment for the environmental covariates is not enough to establish genetic associations. The genetic interpretation of our findings is based on a combination of multiple aspects:

1. No significant influence of the covariates on the parental and sibling effects.
2. Spouses have no significant effects.
3. The associations are additive in the sense that an increase in the number of parents, siblings, or aunts and uncles is associated with an increase in the survival of IPs and the children of IPs. This is a pattern that is not necessarily expected if the findings are due to other, non-genetic, factors that cluster within families (for example wealth).
4. Strong associations between the number of siblings, aunts/uncles and IP/Children of IPs' survival were identified in the presence of non-longevous parents. Furthermore, for aunts and uncles, there is less of a chance of shared environments, which further strengthens the case for genetic influences.
5. The associations between parental, sibling, aunts/uncles longevity, and offspring (either IPs or children of IPs) survival was observed for two consecutive generations in two

databases, covering populations with vastly different environmental circumstances, with remarkably similar effects.

We mentioned the above reasoning for genetic associations in the second paragraph of the discussion (the fourth in the new version of the paper). In the new version we elaborated on our argumentation in line with the above mentioned reasoning.

We also included supplementary table 11 and 12 showing 4 subsequent models to indicate the separate parent and sibling effects in combination with all covariates to show how this leads to the final model. The different models are: 1. Parent association with IP survival, 2. Sibling association with IP survival, 3. Parent and sibling association with IP survival, 4. Parent, sibling, and covariates association with IP survival.

Implementations

We elaborated on our argumentation in line with the above mentioned reasoning and included supplementary table 11 and 12.

Chapter 3: Discussion

paragraph 4 (line 299-314)

Longevity was transmitted even if parents themselves did not become longevous, which supports the notion that a beneficial genetic component was transmitted. Likewise, the identified associations are additive in the sense that an increase in the number of parents, siblings, or aunts and uncles is associated with an increase in the survival of IPs and the children of IPs. This additive pattern is not necessarily expected if the findings are due to other, non-genetic, factors that cluster within families (for example wealth). This evidence is strengthened by the fact that similar additive associations were identified for IPs and children of IPs without longevous parents but with longevous siblings or aunts and uncles (where the latter generally share less environmental influences with the IPs). Further evidence for the transmission of a genetic component was shown by the fact that none of the tested environmental confounders affected the associations between parental/sibling longevity and IP/children survival, as will be discussed further on in the discussion section. In addition, the fact that we observed very similar results between the two databases, which cover populations with vastly different environmentally related mortality regimes, significantly adds to the generalizability of our observations regarding the associations between parental/sibling longevity and IP (F2) and children (F3) survival.

Chapter 2: Results

paragraph 8 (line 197-203)

The survival advantage of IPs with 1 and 2 or more, or exactly 2 top 10% siblings and parents respectively is independent of covariates such as sibship size and religion (LDS church affiliation, Table 2 and Supplementary Table 11 and 12). Religious IPs from Utah have a lower HR than non-religious persons ($HR_{UPDB}=0.73$ (95% CI=0.65-0.81)) and in the UPDB we observe that sibship size has a small influence on the survival of IPs ($HR_{UPDB}=1.01$ (95% CI=1.00-1.02)) whereas in LINKS sibship size has no significant effect ($HR_{LINKS}=1.01$ (95% CI=1.00-1.02)).

Comment G

There is a typo in $N_LINKS = 202,343$ on line 98.

Response

We thank the reviewer for identifying this typo.

Implementations

We have reviewed the entire paper, figures, and tables in order limit the number of editorial mistakes, in line with comment F3 of reviewer 1. We refer to the new manuscript for all adjustments.

Reviewer 3

This submission is an excellent paper with results of wide interest to demographers. It probes two rich historical datasets with families linked across three generations spanning two centuries and brings the data to bear on heritable, potentially genetic, influences on longevity. The analysis is thoroughly documented and wide-ranging. I found that questions that came to mind while reading the paper were frequently addressed later in the text. I recommend this paper for publication.

Comment A

Is the interval in the Cox model being measured from age zero, out to death or censoring, or from some other starting point? Since each Index Person (IP) has to have a child in order to enter the sample, presence in the sample is conditional on survival to some varying age. A word about how left-truncation is being handled would be helpful.

Response

We thank the reviewer for this important remark. Every IP is included in the analyses with a starting age that is either the age at which the IP gave birth to his/her first child or the age at marriage, whichever was the latest. This is because the IP could not have died before this age due to direct selection. The starting point is thus individualized for each IP. Although we mentioned briefly in the methods that we adjusted for left truncation in the IPs, we agree with the reviewer that this was not clearly stated in the text and we have added a more detailed description to the methods section.

Implementations

We adjusted the methods so that it is now clearly explained how adjustment for left and right censoring was made for the IPs and the other groups that are analyzed (spouses, and children of IPs).

Chapter 4: Methods

paragraph 6 (line 544-549)

In all Cox regression analyses, based on expression 1, we accounted for the fact that IPs were selected to have a spouse and at least one child (left truncation) by using an IP specific age at entry in the study based on the IP's age at first child or the age at marriage, whichever was later. A similar approach was followed for the spouses of the IPs and no adjustment for left truncation was necessary for the children of the IPs, since they were not selected in any way. Moreover, we accounted for right censoring in all relatives of the IPs.s

Comment B

From the point of view of genetic demography, survival in the face of senescent mortality is of primary interest. An alternative to the authors' analysis would have been to restrict the sample to IPs over, say, 45, and start the Cox analysis at age 45. For these populations, maternal mortality would seem to be a complicating factor that might be good to avoid. This is not a request for the authors to do a different Cox fitting but instead to explain in more detail the reasons behind their choice.

Response

We thank the reviewer for this constructive remark and would like to discuss and elaborate on our considerations regarding a potential age selection for the IPs. Our primary aim was to utilize the study populations represented in the two databases with as little selections as possible. Many previous studies focused on specific groups, such as centenarians, or nonagenarians. As a result of these selections these studies can only provide a fragmented and potentially selective representation of reality. As a result, it is always unclear to what extent the findings generalize to the general population. The same goes for our recent findings in the Leiden Longevity Study¹² which was one of our inspirations to impose as little selections as possible to the two datasets in this study. By confining the IPs to a certain age we would potentially face this same issue that we wanted to avoid. However, in intergenerational research it is impossible to avoid some selection because in order for multiple generations to exist, a person had to have had children and a spouse. In this light, one of the benefits of replicating our analyses in the children of the IPs is that this group was not subject to any direct selection effects.

In addition, it is possible that a reduced maternal mortality pattern is part of the characteristics that result from the genetic makeup in longevous families. In this light we would actually like to capture maternal mortality and it might even be an interesting trait to investigate in a follow-up study. Apart from this, by excluding IPs based on avoiding maternal mortality we would unnecessarily decrease the size of our study population.

Implementations

No adjustments have been made.

Comment C

The cohort-by-cohort levels of infant mortality in the Swedish lifetables can hardly be expected to capture those in the population in the Utah database, nor those in Zeeland over the centuries. An alternative would have been to calculate percentiles based on lifetables conditional on survival to, say, age 10 or 15. Again, an explanation of the authors' choice would be valuable.

Response

We thank the reviewer for this comment which extends the previous remark. We agree with the reviewer that the Swedish lifetables will likely not capture the levels of infant mortality in Utah or Zeeland. This potential inability to fully capture childhood mortality extends to the Dutch, and other lifetables. Childhood mortality in Zeeland was for example high compared to the general pattern of childhood mortality in the Netherlands²⁴⁻²⁶. So, using lifetables to capture childhood mortality is not perfect but it provides the best way we have to normalize our results for cohort effects and allow equal comparisons between studies^{1,23}.

We chose not to calculate the percentiles conditional on survival to an age that excludes childhood mortality because, as explained above in comment B, we wanted to impose as few selections on the data as possible. The choice to include early life mortality thus influenced us to calculate the percentiles from birth, since we felt it inappropriate to calculate a survival percentile, for example conditional on survival to age 10 if the person could die from age 0. This left us with a similar decision for survival percentile calculation for the IPs, since they could not die before the birth of at least one child or marriage (similar situation for parents of IPs and spouses of IPs). Here we choose to also calculate the survival percentiles without

conditioning because this would make the survival percentiles incomparable to the other groups.

Implementations

No adjustments have been made.

Comment D

The translation between percentiles and ages cohort-by-cohort in Supplementary Figure 1 is quite informative. A few examples of pairings of percentile and age from this figure might be mentioned in the main text. These are apparently the percentile-age pairings from the Swedish lifetables used to calculate percentiles in both datasets, Please say so. Is there some way to give the reader a sense of what the empirical percentiles for the IPs in the two datasets look like, subject to conditioning on survival to some fixed age?

Response

We thank the reviewer for his positive remark about supplementary Figure 1 and we agree that it is important that the reader gets a better intuition of how the percentiles relate to actual ages. The reviewer's remark consists of three parts which we will address separately.

1. The reviewer is correct that supplementary Figure 1 represents the the percentile-age pairings from the Swedish lifetables used to calculate percentiles in both datasets.
2. We agree with the reviewer that it is a good idea to mention a few pairing examples of percentiles and ages in the study.
3. We assume the reviewer would like to have a version of supplementary figure 1 based on percentiles with a specific age cut-off (e.g. top 1, 5, or 10% survivors beyond 15 or 50 years). We added here a figure, similar to supplementary figure 1 conditioned on age 15 and 50.

However, if the reviewer suggests that we show for example if a person belonged to the top 10% survivors of his/her birth cohort based on the UPDB and LINKS data itself, instead of using the Swedish lifetables we can unfortunately not properly do this. Even though the UPDB and LINKS data are large, there is not enough mortality information per birth cohort to derive stable percentiles in a comparable way to what we can do when using lifetables. If the reviewer had another approach than the one we have surmised, we would gladly explore such alternatives.

-This figure represents the the percentile-age pairings from the Swedish lifetables used to calculate survival percentiles in both the UPDB and LINKS datasets.

-Line colors:

- Blue: men
- Red: women

-Line patterns:

- Dotted lines represent the top 1% survivors of the specific birth cohorts
- Broken lines represent the top 5% survivors of the specific birth cohorts
- Unbroken lines represent the top 10% survivors of the specific birth cohorts

Implementations

We now mention in the methods that supplementary figure 1 is based on pairings between ages in the UPDB and percentiles corresponding to that age based on the Swedish lifetables. We further mention some of the pairings in the text.

Chapter 4: Methods

Paragraph 6 (line 573-478)

Supplementary Figure 1 shows the ages at death corresponding to the top 10, 5, and 1 percent survivors for the UPDB and LINKS. This figure can be used to map the percentiles, which are based on percentile-age pairings from the Swedish lifetables, to absolute ages. For example: a top 10% female in 1750 matched an age of 76 years whereas this was 74 years for males. In 1850 a top 10% female and male matched an age of 83 years and 81 years respectively.

Comment E

The labeling on the Y-axis for the right-hand panels in Figure 2 is misleading. The words 'cumulative hazard' should appear in the label, not just in the caption. These panels would be more informative if the Y-axis only reached out to, 1.5 or 2.0 or so and if the X-axis were cut off at 90 or so. In the present version, the range is too broad and the resolution too small for the curves to be easily distinguished by eye.

Response

We agree with the reviewer that the figures would benefit from a label description.

We also agree with the reviewer that the cumulative hazard figures are not ideal to show the difference between the cumulative hazard lines for the early ages. As the reviewer suggested, we restricted the Y-axis at 3.0 and as a function of that, the x-axis only contain information to survival at 90. We feel that in this way we do not use the cumulative hazard plot to its full potential, which for the purpose of this paper is that we show a survival advantage over the life course up to age 100 and to show that the proportional hazards assumption in general holds. In order to convey this to the reader we prefer to show the cumulative hazards for the survival to 100.

We provide an example below of how the cumulative hazard plots look like when cutting at 3.0 on the y-axis and we hope that the reviewer agrees with us to leave the cumulative hazards plots the way they are.

Implementations

We adjusted the label description of figure 2 and the related supplementary figures in line with the reviewer's suggestion.

Chapter 2: Results

Figure 2

-This figure depicts the Hazard Ratio (HR) for IPs (left column) with 1 and 2 parents or 1 and 2+ siblings belonging to the top x percentile ($x = 1, 2, 3, \dots, 60$) of survivors of their birth cohort. The percentile groups (x -axis) are mutually inclusive, meaning that a first degree family member who belonged to the top 1% also belonged to the top 5% etc. The figure also depicts the Cumulative Hazard (CH) for IPs (right column) with 1 and 2 parents or 1 and 2+ siblings who belong to the top 10%.

-Green (dotted) lines present the reference group of 0 top x percentile parents or siblings, yellow lines represent 1 top x percentile parents or siblings, blue lines represent 2 or 2+ top x percentile siblings.

-left column: x-axes represent the top x birth cohort based survival percentile, the y-axes represent the hazard ratio (HR) of dying for IPs having 1 and 2 or 2+ top x percentile parents or siblings compared to having 0 top x percentile parents or siblings.

-right column: x-axes represent IP years of survival, y-axes represent the IPs' cumulative hazard of dying while having 1 and 2 or 2+ top 10th percentile parents or siblings compared to having 0 top 10th percentile parents or siblings.

-All estimates are adjusted for religion (UPDB only), sibship size, birth cohort, sex, socio-economic status, mother's age at birth, birth order, birth intervals, twin birth, and number of top 10% parents or number of top 10% siblings for the sibling and parent analyses respectively.

Comment F

More discussion of the geographical spread of the IPs and their parents and children in the ``Utah' database would be welcome. The database was originally built around Utah founder families. But is it not true that many or most of the individuals who appear in the database were not living their lives in Utah?

Response

Indeed the geographical spread of the IPs and their relatives is an interesting point. Utah was first settled in 1847 which means that some of the UPDB persons may not have lived in Utah. We extended the description of the populations in the methods section with information about the geographical spread of the persons from the UPDB included in this study and added an overview table (supplementary table 13), with information about the geographical spread for the persons included in this study, to the supplementary material.

Implementations

We added a paragraph to the data description in the methods section describing the proportion of persons who lived in Utah and separate this by group (IP, father, mother, etc.) and birth continent. We also included supplementary table 13, depicting an overview of these numbers.

Chapter 4: Methods

Paragraph 5 (line 444-543)

All individuals in LINKS have at some point in their lives lived in Zeeland, this is because the data were constructed based on vital event records from Zeeland. Utah was first settled in 1847 and in the UPDB mortality information for ancestors of Utah associated persons are available. As a result not all persons necessarily had to live in Utah. Supplementary Table 13 shows that in our data, 97 percent of the IPs lived in Utah. This percentage is lower for their fathers (80%) and mothers (87%), and is an expected pattern given the historic nature of how Utah was settled. Furthermore, 70% of the siblings, 97% of the spouses, and 92% of the children lived in Utah. The majority of the persons from our sample who lived in Utah, migrated from another state in the US to Utah (87%), 12% came from Europe and 1% from the rest of the world.

Chapter 4: Methods

Supplementary table 13

Supplementary table 13: Demographic spread for included UPDB persons

Role	Fathers	Mothers	IPs	Siblings	Spouses	Children
Lived in Utah (%)	7600 (80)	8478 (87)	9901 (97)	32026 (70)	9998 (97)	49839 (92)
Not lived in Utah (%)	1884 (20)	1229 (13)	345 (3)	13674 (30)	356 (3)	4236 (8)

Birth continent	America	Europe	Asia	Africa	Australia	Other
Lived in Utah	85141 (91)	11490 (73)	35 (65)	4 (100)	195 (73)	171 (34)
Not lived in Utah	8310 (9)	4363 (27)	19 (45)	0 (0)	73 (27)	329 (66)

-Numbers are based on uncensored individuals

-Numbers are based on the UPDB only

Comment G

In the Supplementary Information, a histogram of the empirical values of the percentile points for the deaths of the IPs would be welcome, so that readers could compare them to a uniform distribution.

Response

We assume that the reviewer would like to see a histogram illustrating the distribution of percentiles for the Index Persons. We agree with the reviewer that this might be interesting, especially in relation to comment H. Though, the remark goes beyond the original scope of the paper because we measured survival (follow up time) for the IPs as age at death instead of percentiles (methods, paragraph 3 (line 454-461).

This comment made us realize that we have to clarify in the introduction of the paper that we measure survival (follow up time) for the IPs as age at death and for the children of IPs as age at death or age at last observation.

Because we measured IP follow up time as age at death we prefer to show a four panel figure to the reader. The top row shows the age at death distribution for the IPs which underlies the cox models and the bottom row shows the percentile distribution for the IPs.

Implementations

We added to the first paragraph of the results that the distribution of age at death is depicted in supplementary figure 9.

We further introduced the different terminology regarding survival percentiles (which were used as predictors in the cox models) and survival (the outcome the cox models), in terms of follow up time (age in years) in the introduction of the paper.

Chapter 2: Results

Paragraph 1 (line 97-110)

We identify three generations of families in the UPDB and LINKS covering 10,246 and 10,114 families, respectively, who are centered around a single index person (IPs,F2) per family (Figure 1). We identify parents (F1, $N_{UPDB}=20,492$ & $N_{LINKS}=20,228$), siblings (F2, $N_{UPDB}=54,144$ & $N_{LINKS}=53,978$), spouses (F2, $N_{UPDB}=11,230$ & $N_{LINKS}=10,788$), and children (F3, $N_{UPDB}=61,104$ & $N_{LINKS}=62,495$) for all IPs in both datasets (Table 1). IPs are born between 1767 and 1902 in the UPDB, and between 1797 and 1902 in LINKS. In the UPDB, 51% of the IPs are female, compared to 53% in LINKS. The IPs' mean age at death is 70.88 (SD=16.03) years in the UPDB and 63.86 (SD=17.99) years in LINKS. No IPs are censored, as they are selected to have an available birth and death date. In addition, Supplementary figure 9 shows the age at death distribution for the IPs in both datasets. In the following sections we explore associations between IP survival and the number of 1-60% surviving parents and siblings in a cumulative analysis and subsequently identify in mutually exclusive IP groups the survival percentile threshold that drives the cumulative effect and demarcates longevity (see methods section)

Chapter 1: Introduction

paragraph 3 (line 78-95)

We use the data available in the Utah Population Database (UPDB,Utah) and the LINKing System for historical family reconstruction (LINKS,Zeeland) based on US and Dutch citizens, respectively. Zeeland was a region with difficult living conditions compared to Utah (see methods section). In these datasets we identify 20,360 three-generational families (F1-F3) containing index persons (IPs, F2), their parents (F1), siblings (F2), spouses (F2), and children (F3) comprising 314,819 persons in total. First, we examine the association between the survival, measured as age at death, of IPs (F2) and the number of parents (F1) and siblings (F2) belonging to the top 1-60% of their birth cohort, in a cumulative way (comparing mutually inclusive percentile groups). Second, we determine the survival percentile threshold that drives the cumulative effects as a criterion for defining human longevity by investigating IP (F2) survival when divided into mutually exclusive groups based on the longevity of their parents (F1) and siblings (F2). Third, we focus on the top 10% parents and siblings to investigate whether longevous and non-longevous parents, with increasing numbers of longevous siblings, transmit longevity to the IPs. Fourth, we confirm our findings in the next generation (F3) by examining the association between the survival, measured as age at death or last observation, of IPs' children (F3) and longevity of IPs (F2), their spouses (parents, F2) and siblings (aunts and uncles, F2). Finally, we explore potential environmental influences by studying spouses (F2) of longevous IPs (F2).

Comment H

In the Supplementary Information, it would also be interesting to see a scatterplot of the percentile for the longer-lived parent versus the percentile for the IP. Another option would be a scatterplot of the average of the father's and mother's percentiles versus the percentile for the IP, perhaps subsampled so that the scatter of points is more easily visible. Such plots would make it easier to visualize the (piecewise) parent-child correlations that underlie the Cox Model results, in a form analogous to Galton's famous plot for heights.

Response

We thank the reviewer for this suggestion. We assume that the reviewer is interested to see how much variation there is for the IPs within the categories depicted in figure 3.

As mentioned in comment G, we measured IP follow up time as age at death. We find it very interesting to show the variation based on the IP survival percentile instead of age at death, though it would extend the scope of the paper. We therefore prefer to show a four panel figure, equal to comment G.

We propose to illustrate the IP variation with a boxplot instead of constructing a replication of Galton's plot for heights because a replication of Galton's plot based on age at death or survival percentile would obscure the findings we show in figure 3. This is because the parent - offspring association for lifespan is weak compared to longevity but also compared to height, which is strongly heritable. The proposed boxplot shows the median (plus quantiles and single values) age at death and survival percentiles of the IPs grouped by the 6 parental categories, relating to figure 3. We extend the reviewer's suggestion slightly by adding to what extent top 10% IPs (F2) are represented when grouped by the 6 parental (F1) categories.

-This figure relates to main figure 3 and shows the median + quantiles and variation for IPs' age at death on the top row. The bottom row shows the median + quantiles and variation for IPs' survival percentiles.

-Nodes are based on 1/3th of the total sample size for illustrative purposes.

-The red lines on the bottom row represent the cut-off for the top 10 percent surviving IPs for the different groups. Similar to the decrease in HR for the different groups illustrated in main figure 2 and the increase in age at death or survival percentile for the different groups illustrated in this figure, there is an increase in top 10% surviving IPs. For the UPDB data 17% of the total number of IPs in group 6 belongs to the top 10% survivors, this is 19% for group 5, 23% for group 4, 26% for group 3, 29% for group 2, and 37% for group 1. For the LINKS data the numbers, in similar order, are 13%, 14%, 16%, 18%, 23% and 32%.

Implementations

We included and refer to the above figure as supplementary figure 10 in the results section.

Chapter 2: Results

Paragraph 4 (line141-151)

To determine the survival percentile threshold that drives the survival advantage of IPs (F2) with the number of top 1-60% parents (F1), as shown in Figure 2, we construct 6 mutually exclusive IP (F2) groups (g) based on the survival percentiles of F1 parents (g1= $[\geq 0^{\text{th}} \ \& \ \leq 1^{\text{th}}$ percentile], g2= $[\geq 1^{\text{th}} \ \& \ \leq 5^{\text{th}}$ percentile], g3= $[\geq 5^{\text{th}} \ \& \ \leq 10^{\text{th}}$ percentile], g4= $[\geq 10^{\text{th}} \ \& \ \leq 15^{\text{th}}$ percentile], g5= $[\geq 15^{\text{th}} \ \& \ \leq 20^{\text{th}}$ percentile], g6= $[\geq 20^{\text{th}} \ \& \ \leq 100^{\text{th}}$ percentile], see methods section) and compare groups 1-5 with group 6. Figure 3A and B show the HRs of IP groups for the UPDB and LINKS and is supplemented by the IP age at death and survival percentile variation, as depicted in Supplementary figure 10. Figure 3A and B illustrate that IPs in group 1, 2, 3, and 4 have a significant survival advantage compared to group 6, with the lowest HR for group 1 in both the UPDB and LINKS ($HR_{\text{max-UPDB}}=0.76$ (95% CI=0.67-0.86) and $HR_{\text{max-LINKS}}=0.72$ (95% CI=0.60-0.86)). Group 5 does not statistically differ from group 6 ($HR_{\text{group5-UPDB}}=1$ (95% CI=0.91-1.10) and $HR_{\text{group5-LINKS}}=0.96$ (95% CI=0.87-1.05)) and thus, these effects indicate that the top 15% surviving parents drive the association with the survival advantage of IPs as shown in Figure 2.

References

1. Sebastiani, P., Nussbaum, L., Andersen, S. L., Black, M. J. & Perls, T. T. Increasing Sibling Relative Risk of Survival to Older and Older Ages and the Importance of Precise Definitions of 'Aging,' 'Life Span,' and 'Longevity'. *Journals Gerontol. Ser. A Biol. Sci. Med. Sci.* **00**, 1–7 (2015).
2. Gavrilova N.S., G. L. A., Gavrilov, L. A. & Gavrilova, N. S. When does human longevity start?: Demarcation of the boundaries for human longevity. *Rejuvenation Res.* **4**, 115–124 (2001).
3. Ljungquist, B., Berg, S., Lanke, J., McClearn, G. E. & Pedersen, N. L. The Effect of Genetic Factors for Longevity : A Comparison of Identical and Fraternal Twins in the Swedish Twin Registry. *J. Gerontol. Med. Sci.* **53A**, 441–446 (1998).
4. Gögele, M. *et al.* Heritability analysis of life span in a semi-isolated population followed across four centuries reveals the presence of pleiotropy between life span and reproduction. *J. Gerontol. A. Biol. Sci. Med. Sci.* **66**, 26–37 (2011).
5. Perls, T. T. *et al.* Life-long sustained mortality advantage of siblings of centenarians. *Proc. Natl. Acad. Sci.* **99**, 8442–8447 (2002).
6. Kerber, R. A., Brien, E. O., Smith, K. R. & Cawthon, R. M. Familial Excess Longevity in Utah Genealogies. *Journals Gerontol. Ser. A Biol. Sci. Med. Sci.* **56**, 130–139 (2001).
7. Gudmundsson, H., Gudbjartsson, D. F. & Kong, A. Inheritance of human longevity in Iceland. *Eur. J. Hum. Genet. EJHG* **8**, 743–749 (2000).
8. Schoenmaker, M. *et al.* Evidence of genetic enrichment for exceptional survival using a family approach: the Leiden Longevity Study. *Eur. J. Hum. Genet.* **14**, 79–84 (2006).
9. Hjelmborg, J. vB *et al.* Genetic influence on human lifespan and longevity. *Hum. Genet.* **119**, 312–21 (2006).
10. Willcox, B. J., Willcox, D. C., He, Q., Curb, J. D. & Suzuki, M. Siblings of Okinawan centenarians share lifelong mortality advantages. *J. Gerontol. A. Biol. Sci. Med. Sci.* **61**, 345–54 (2006).
11. Pedersen, J. K. *et al.* The Survival of Spouses Marrying Into Longevity-Enriched Families. *Journals Gerontol. Ser. A Biol. Sci. Med. Sci.* **72**, 1–6 (2017).
12. Berg, N. van den *et al.* Longevity Around the Turn of the 20th Century: Life-Long Sustained Survival Advantage for Parents of Today's Nonagenarians. *Journals Gerontol. Ser. A* **73**, 1295–1302 (2018).
13. Perls, T. T., Bubrick, E., Wager, C. G., Vijg, J. & Kruglyak, L. Siblings of centenarians live longer. *Lancet* **351**, 1560 (1998).
14. Houde, L., Tremblay, M. & Vézina, H. Intergenerational and Genealogical Approaches for the Study of Longevity in the Saguenay-Lac-St-Jean Population. *Hum. Nat.* **19**, 70–86 (2008).
15. Kemkes-Grottenthaler, A. Parental effects on offspring longevity—evidence from 17th to 19th century reproductive histories. *Ann. Hum. Biol.* **31**, 139–158 (2004).
16. Deluty, J. A., Atzmon, G., Crandall, J., Barzilai, N. & Milman, S. The influence of gender on inheritance of exceptional longevity. *Aging (Albany. NY).* **7**, 412–418 (2015).
17. You, D., Gu, D. & Yi, Z. Familial Transmission of Human Longevity Among the Oldest-Old in China. *J. Appl. Gerontol.* **29**, 308–332 (2009).
18. Gavrilov, L. A. & Gavrilova, N. S. *Predictors of Exceptional Longevity: Effects of Early-Life and Midlife Conditions, and Familial Longevity.* **19**, (2015).

19. Wilmoth, J. R. & Shkolnikov, V. Human Mortality Database. *University of California, Berkeley (USA), and Max Planck Institute for Demographic Research (Germany)* (2017).
20. Van Der Meulen, A. *Life tables and Survival analysis*. (2012). at <<https://www.cbs.nl/NR/rdonlyres/C047245B-B20E-492D-A411-9F298DE7930C/0/2012LifetablesandSurvivalanalysart.pdf>>
21. Carolina, T., Uijenhoven, L. & van der Laan, J. Overlevingstafels en longitudinale analyse. *CBS* 1–25 (2009). at <<https://www.cbs.nl/NR/rdonlyres/98267DE5-D866-4AA9-8E16-25FA57A90C3B/0/2010x3710pub.pdf>>
22. Lundstrom, H. *Cohort mortality in Sweden: Mortality statistics since 1861*. (2010).
23. van den Berg, N., Beekman, M., Smith, K. R., Janssens, A. & Slagboom, P. E. Historical demography and longevity genetics: Back to the future. *Ageing Res. Rev.* **38**, 28–39 (2017).
24. van Poppel, F., Jonger, M. & Mandemakers, K. Differential infant and child mortality in three Dutch regions, 1812–1909. *Econ. Hist. Rev.* **58**, 1–38 (2005).
25. Hoogerhuis, O. W. *Baren op Beveland. Vruchtbaarheid en zuigelingensterfte in Goes en omliggende dorpen gedurende de 19e eeuw*. (AAG Bijdragen 42, 2003).
26. van dijk, I. K. & Mandemakers, K. Historical Life Course Studies Like Mother, Like Daughter. Intergenerational Transmission of Infant Mortality Clustering in Zeeland, the Netherlands, 1833-1912. *Hist. Life Course Stud.* 1–26 (2018).

Reviewer #1 (Remarks to the Author):

The authors addressed the review comments in a thorough manner. I would suggest to discuss the paper recently published by CalicoLab (Ruby et al., Genetics), which argues that genetic component of lifespan is negligible and heritability of lifespan is explained mainly by the assortative mating. Also, the non-linear pattern of heritability of lifespan was first shown in the following paper: Gavrilova et al. Evolution, mutations and human longevity. Human Biology, 1998, 70(4): 799-804. Minor issue: I suggest to check the list of references. For example in reference 50 the journal title is missing.

Reviewer #2 (Remarks to the Author):

My criticisms have been adequately addressed, and I think the paper is well worth publishing as is. I noticed two typos, in l. 529 (missing %, I think) and l. 566 (missing =).

Reviewer #3 (Remarks to the Author):

The authors have been extremely responsive to all the original reviews, adding a considerable amount of new discussion, references to the literature, and additional tables and figures. They have clarified many details of their methods and improved the labeling of key figures. Their Response to Reviews is full of thoughtful explanations of the decisions they have made in revision. In my view, all points in the reviews have been addressed in a satisfactory fashion. I appreciate all the time and effort that the authors have put in for this revision, exceeding what is usually seen.

Reviewer 1

The authors addressed the review comments in a thorough manner.

Comment A

I would suggest to discuss the paper recently published by CalicoLab (Ruby et al., Genetics), which argues that genetic component of lifespan is negligible and heritability of lifespan is explained mainly by the assortative mating.

Response

We agree with the reviewer that it is important to discuss the recent paper by Ruby et al. Discussing this paper provides a good opportunity to elaborate on the differences between the phenotypes of lifespan and longevity, as those two phenotypes are often confused, and stretch the importance of the longevity phenotype for identification of genetic loci. We also understand that their conclusions will need to be verified, especially with more direct measures of features on which the assortative mating is based.

Implementations

We added a section to the final paragraph of the discussion where we address the low heritability of lifespan and the strong familial clustering of longevity in the light of identifying genetic longevity loci.

Chapter 3: Discussion

Paragraph 8 (line 371-393)

Human Lifespan (defined as age at death) has a low heritability in the population at large¹⁻⁴. Studies estimated the heritability of lifespan between 12 and 25%¹⁻³ and a recent study estimated that the heritability of lifespan was even lower, ~7%, after adjustment for the lifespans of nongenetic (in-law) relatives⁴. Therefore lifespan based gene mapping may not be fruitful. In addition, the genetic component of lifespan includes the heritability of early life mortality, which is mainly due to disease and external causes. Despite the low heritability and polygenic architecture^{5,6} of lifespan, recent genetic studies have identified^{7,8} and replicated⁹ some lifespan loci of which the rare alleles lower the risk of age-related diseases. Hence, using the lifespan trait hampers the identification of genetic loci contributing to survival into extreme ages (longevity). Longevity however, clusters strongly within families as shown by previous studies¹⁰⁻¹⁴ and robustly quantified in this study. Hence, the longevity trait is much more promising and appropriate for the identification of genetic loci contributing to survival into extreme ages and should not be confused with the lifespan trait^{1,15}. Our results imply that to find loci that promote survival to the highest ages in the population, genetic studies should be based on long-lived cases including at least parental mortality information but preferably also mortality information of siblings and other first and second degree relatives. The longevity threshold should include cases belonging up to the top 10% survivors, with parents belonging up to the top 15% survivors of their birth cohort and siblings belonging up to the top 10% survivors of their birth cohort. To sharpen the longevity effect, the percentile threshold applied may be made more extreme but would likely lead unnecessarily to a sample size with limited power. If our proposed longevity

definition is consistently applied across studies the comparative nature of longevity studies may improve and facilitate the discovery of novel genetic variants.

Comment B

the non-linear pattern of heritability of lifespan was first shown in the following paper: Gavrilova et al. Evolution, mutations and human longevity. Human Biology, 1998, 70(4): 799-804.

Response

We thank the reviewer for pointing out the paper by Gavrilova et al. We now refer to the paper when discussing the previous literature on the non-linear pattern of the heritability of lifespan. Furthermore, to avoid confusion about who was the very first to identify the different aspects mentioned in paragraph 2 of the discussion, we no longer say “were the first”.

Implementations

We included the paper by Gavrilova et al. and deleted the text mentioning “were the first” in paragraph 2 of the discussion.

Chapter 3: Discussion

paragraph 2 (line 276-285)

Previous studies of smaller sample size than the current study, usually focusing on two generations of selected data (for mortality or geographical locations) identified (1) an increase in the heritability of lifespan with parental age^{13,16,17} and showed high recurrence risks between parental and offspring or sibling longevity. Thus, providing indications that the heritability of longevity may be stronger than that of lifespan^{10,12,18-20}, (2) that sibling relative risks beyond the top 5% survivors might not increase in a linear fashion¹⁵ and that this non-linearity may indicate the existence of a longevity threshold²¹, and (3) longevity recurrence risks for siblings or parents of selected longevous individuals^{10-12,14,15,22,23} and showed increased survival probabilities and longevity recurrence risks for children of longevous parents^{12,18,19,24-26}.

Comment C

I suggest to check the list of references. For example in reference 50 the journal title is missing.

Response

We thank the reviewer for pointing out some mistakes in the reference list. We checked all references and adjusted mistakes when identified.

Implementations

We checked all references and adjusted mistakes when identified.

Reviewer 2

My criticisms have been adequately addressed, and I think the paper is well worth publishing as is.

Comment A

I noticed two typos, in l. 529 (missing %, I think) and l. 566 (missing =).

Response

We thank the reviewer for pointing out these typos.

Implementations

We adjusted the typos.

Chapter 4: Methods

paragraph 5 (line 538-541)

Based on the results presented in the cumulative and mutually exclusive group analyses we focused on the top 10% surviving family members because the mutually exclusive group analysis (analysis 2) indicated longevity effects for siblings beyond the top 10% and 15% for siblings and parents respectively.

Chapter 4: Methods

paragraph 2 (line 575-576)

In analysis 3, based on the top 10%, the IP will contribute with a weight of $w=3/4$. In this way we avoid a potential advantage of larger families to be represented in more extreme groups.

Reviewer 3

The authors have been extremely responsive to all the original reviews, adding a considerable amount of new discussion, references to the literature, and additional tables and figures. They have clarified many details of their methods and improved the labeling of key figures. Their Response to Reviews is full of thoughtful explanations of the decisions they have made in revision. In my view, all points in the reviews have been addressed in a satisfactory fashion. I appreciate all the time and effort that the authors have put in for this revision, exceeding what is usually seen.

Response

We thank the reviewer for this positive remark. We enjoyed reading and incorporating all the positive input of the reviewer (and the other reviewers) and believe that the quality of the paper is significantly improved by all the reviewers' remarks.

References

1. van den Berg, N., Beekman, M., Smith, K. R., Janssens, A. & Slagboom, P. E. Historical demography and longevity genetics: Back to the future. *Ageing Res. Rev.* **38**, 28–39 (2017).
2. Herskind, A. M. *et al.* The heritability of human longevity: A population-based study of 2872 Danish twin pairs born 1870–1900. *Hum. Genet.* **97**, 319–323 (1996).
3. Kaplanis, J. *et al.* Quantitative analysis of population-scale family trees with millions of relatives. *Science* **360**, 171–175 (2018).
4. Ruby, J. G. *et al.* Estimates of the Heritability of Human Longevity Are Substantially Inflated due to Assortative Mating. *Genetics* **210**, 1109–1124 (2018).
5. Shadyab, A. H. & LaCroix, A. Z. Genetic factors associated with longevity: A review of recent findings. *Ageing Res. Rev.* **19**, 1–7 (2015).
6. Christensen, K., Johnson, T. E. & Vaupel, J. W. The quest for genetic determinants of human longevity: challenges and insights. *Nat. Rev. Genet.* **7**, 436–48 (2006).
7. Pilling, L. C. *et al.* Human longevity: 25 genetic loci associated in 389,166 UK biobank participants. *Aging (Albany, NY)*. **9**, 2504–2520 (2017).
8. Joshi, P. K. *et al.* Genome-wide meta-analysis associates HLA-DQA1/DRB1 and LPA and lifestyle factors with human longevity. *Nat. Commun.* **8**, 1–13 (2017).
9. Timmers, P. R. H. J. *et al.* Genomic underpinnings of lifespan allow prediction and reveal basis in modern risks. *bioRxiv* 1–71 (2018). at <http://biorxiv.org/content/early/2018/07/06/363036.abstract>
10. Perls, T. T. *et al.* Life-long sustained mortality advantage of siblings of centenarians. *Proc. Natl. Acad. Sci.* **99**, 8442–8447 (2002).
11. Pedersen, J. K. *et al.* The Survival of Spouses Marrying Into Longevity-Enriched Families. *Journals Gerontol. Ser. A Biol. Sci. Med. Sci.* **72**, 109–114 (2017).
12. Schoenmaker, M. *et al.* Evidence of genetic enrichment for exceptional survival using a family approach: the Leiden Longevity Study. *Eur. J. Hum. Genet.* **14**, 79–84 (2006).
13. Ljungquist, B., Berg, S., Lanke, J., McClearn, G. E. & Pedersen, N. L. The Effect of Genetic Factors for Longevity: A Comparison of Identical and Fraternal Twins in the Swedish Twin Registry. *Journals Gerontol. Ser. A Biol. Sci. Med. Sci.* **53A**, 441–446 (1998).
14. Berg, N. van den *et al.* Longevity Around the Turn of the 20th Century: Life-Long Sustained Survival Advantage for Parents of Today’s Nonagenarians. *Journals Gerontol. Ser. A* **73**, 1295–1302 (2018).
15. Sebastiani, P., Nussbaum, L., Andersen, S. L., Black, M. J. & Perls, T. T. Increasing Sibling Relative Risk of Survival to Older and Older Ages and the Importance of Precise Definitions of “Aging,” “Life Span,” and “Longevity”. *Journals Gerontol. Ser. A Biol. Sci. Med. Sci.* **71**, 340–346 (2016).
16. Gögele, M. *et al.* Heritability analysis of life span in a semi-isolated population followed across four centuries reveals the presence of pleiotropy between life span and reproduction. *J. Gerontol. A. Biol. Sci. Med. Sci.* **66**, 26–37 (2011).
17. Gavrilova, N. *et al.* Evolution, mutations, and human longevity European royal and noble families. *Hum. Biol.* **70**, 799–804 (1998).

18. Kerber, R. A., Brien, E. O., Smith, K. R. & Cawthon, R. M. Familial Excess Longevity in Utah Genealogies. *Journals Gerontol. Ser. A Biol. Sci. Med. Sci.* **56**, 130–139 (2001).
19. Gudmundsson, H., Gudbjartsson, D. F. & Kong, A. Inheritance of human longevity in Iceland. *Eur. J. Hum. Genet. EJHG* **8**, 743–749 (2000).
20. Hjelmborg, J. vB *et al.* Genetic influence on human lifespan and longevity. *Hum. Genet.* **119**, 312–321 (2006).
21. Gavrilova N.S., G. L. A., Gavrilov, L. A. & Gavrilova, N. S. When does human longevity start?: Demarcation of the boundaries for human longevity. *Rejuvenation Res.* **4**, 115–124 (2001).
22. Willcox, B. J., Willcox, D. C., He, Q., Curb, J. D. & Suzuki, M. Siblings of Okinawan centenarians share lifelong mortality advantages. *J. Gerontol. A. Biol. Sci. Med. Sci.* **61**, 345–54 (2006).
23. Perls, T. T., Bubrick, E., Wager, C. G., Vijg, J. & Kruglyak, L. Siblings of centenarians live longer. *Lancet* **351**, 1560 (1998).
24. Houde, L., Tremblay, M. & Vézina, H. Intergenerational and Genealogical Approaches for the Study of Longevity in the Saguenay-Lac-St-Jean Population. *Hum. Nat.* **19**, 70–86 (2008).
25. Kemkes-Grottenthaler, A. Parental effects on offspring longevity—evidence from 17th to 19th century reproductive histories. *Ann. Hum. Biol.* **31**, 139–158 (2004).
26. Deluty, J. A., Atzmon, G., Crandall, J., Barzilai, N. & Milman, S. The influence of gender on inheritance of exceptional longevity. *Aging (Albany. NY).* **7**, 412–418 (2015).